# Embryonic keratin19+ progenitors generate multiple functionally distinct progeny to maintain epithelial diversity in the adult thymus medulla

Beth Lucas [1], Andrea J. White[1], Fabian Klein [2], Clara Veiga-Villauriz[2], Adam Handel [2,3], Andrea Bacon[1], Emilie J. Cosway[1], Kieran D. James[1], Sonia M. Parnell[1], Izumi Ohigashi [4], Yousuke Takahama [5], William E. Jenkinson[1], Georg A. Hollander [2,6,7], Wei-Yu Lu[8] & Graham Anderson [1] ✉

The thymus medulla is a key site for immunoregulation and tolerance, and its functional specialisation is achieved through the complexity of medullary thymic epithelial cells (mTEC). While the importance of the medulla for thymus function is clear, the production and maintenance of mTEC diversity remains poorly understood. Here, using ontogenetic and inducible fate-mapping approaches, we identify mTEC-restricted progenitors as a cytokeratin19+ (K19+) TEC subset that emerges in the embryonic thymus. Importantly, labelling of a single cohort of K19+ TEC during embryogenesis sustains the production of multiple mTEC subsets into adulthood, including CCL21+ mTEC[lo], Aire+ mTEC[hi] and thymic tuft cells. We show K19+ progenitors arise prior to the acquisition of multiple mTEC-defining features including RANK and CCL21 and are generated independently of the key mTEC regulator, Relb. In conclusion, we identify and define a multipotent mTEC progenitor that emerges during embryogenesis to support mTEC diversity into adult life.

The thymus is a primary lymphoid organ uniquely specialised for the production of T-cells. During intrathymic T-cell development, thymic epithelial cells (TEC) play multiple critical roles, and this is reflected by the existence of multiple TEC subsets that are phenotypically and functionally distinct[1,2]. Although intrathymic location identifies anatomically distinct cortical and medullary TEC (cTEC, mTEC) subsets, understanding of the functional importance of TEC diversity is perhaps most evident from studies on the thymic medulla. Consequently,

the thymus medulla is known for its ability to regulate both innate and adaptive components of the immune system. For adaptive immunity, CCL21+ cells within MHCII[lo]CD80[lo] mTEC (mTEC[lo]) control the entry of newly produced single positive CD4+ and CD8+ thymocytes into the thymus medulla[3,4] where T-cell tolerance mechanisms take place. Here, Aire-expressing MHCII[hi]CD80[hi] mTEC (mTEC[hi]) shape the TCR repertoire by mediating the negative selection of potentially autoreactive cells and supporting the lineage divergence of CD4+ thymocytes that

[1]Institute of Immunology and Immunotherapy, University of Birmingham, Birmingham B15 2TT, UK. [2]Department of Paediatrics and Institute of Developmental and Regenerative Medicine, University of Oxford, Oxford, UK. [3]Nuffield Department of Clinical Neurosciences, University of Oxford, Oxford, UK. [4]Institute for Advanced Medical Sciences, Tokushima University, Tokushima, Japan. [5]Thymus Biology Section, Experimental Immunology Branch, NCI/NIH, Bethesda, USA. [6]Paediatric Immunology, Department of Biomedicine, University of Basel and University Children's Hospital Basel, Basel, Switzerland. [7]Department of Biosystems Science and Engineering, ETH Zurich, Basel, Switzerland. [8]Centre for Inflammation Research, Queen's Medical Research Institute, University of Edinburgh, Edinburgh, UK. ✉e-mail: g.anderson@bham.ac.uk

generates immunoregulatory Foxp3[+] T-cells[5–7]. For innate immune cells, Relb-dependent mTEC development is essential for intrathymic production of CD1d-restricted iNKT-cells that regulate the intrathymic dendritic cell (DC) pool[8–10]. In addition, IL25-producing DCLK1[+] thymic tuft cells that reside within the medulla regulate multiple events including T-cell tolerance[11] and control of intrathymic ILC and NKT populations[9,12]. Thus, mTEC diversity supports the generation and maintenance of multiple innate and adaptive immune components. Despite this, while our understanding of mTEC heterogeneity is becoming clearer[11–15], the mechanisms that support the emergence and persistence of multiple mTEC compartments that are important for thymus function remain unclear.

Here, we studied the embryonic thymus and searched for cells that might support and sustain cellular diversity within mTEC. We describe the presence of a cytokeratin 19[+] (K19[+]) TEC subset that is first detectable at E12.5 of gestation, and peaks in frequency in time with the appearance of organised medullary areas. We show K19[+] cells arise independently of the key mTEC regulator, Relb, and lack defining mTEC features including RANK, CCL21 and Aire. Inducible fate-mapping analysis shows that embryonic K19[+] cells give rise to multiple functionally distinct mTEC subsets, including Aire[+] mTEC[hi], CCL21[+] mTEC and DCLK1[+] tuft cells. Moreover, labelling of a cohort of K19[+] cells on a single gestational day is sufficient for the maintenance of mTEC diversity into adulthood. Taken together, our studies define a multipotent mTEC progenitor that arises during embryonic life and supports the long-term generation of mTEC diversity required to sustain thymus medulla function.

## Results

### A transient K19[+] TEC subset emerges during thymus ontogeny

Current models of cTEC and mTEC development describe multiple progenitor populations, including bipotent progenitors which are followed in development by the emergence of cTEC and mTEC lineage-restricted progenitors[16–21]. However, previous attempts to define TEC progenitors and their developmental potential could not take into account newly discovered TEC complexity revealed via RNA sequencing (RNAseq) analysis[11–14]. As such, the identification of TEC progenitors, and examination of their developmental potential, remains an important yet poorly understood area of thymus biology.

To examine this, we performed ontogenetic analysis of the TEC compartment in the embryonic thymus using flow cytometry and confocal microscopy. We began at E12.5, where defined cortex and medulla areas are lacking, and ended at E17.5 where cortex/medulla compartmentalisation is evident[22,23]. This allowed investigation of the sequential development of cTEC and mTEC during the first stages of cortex/medulla formation. Given that cytokeratin-19 (K19) expression has been shown to be a marker of hepatic progenitor cells[24,25], together with the shared endodermal origin of liver and thymus, we focussed efforts on this marker and hypothesised that analysis of K19 expression may help identify potential new markers of TEC progenitors. Interestingly, flow cytometric analysis revealed an EpCAM1[+]K19[+] TEC population that was evident throughout the E12.5–E17.5 period of gestation (Fig. 1a). Consistent with this period being an important phase of thymus growth, the numbers of EpCAM1[+] total TEC and EpCAM1[+]K19[-] TEC were increased as embryogenesis progressed (Fig. 1b, c). In contrast, proportions and numbers of K19[+] TEC did not follow the same pattern, and instead peaked around E14.5–E15.5 (Fig. 1d) and declined by E17.5 (Fig. 1d). Interestingly, our analysis of RNAseq data from Magaletta et al.[26] shows that K19 is expressed in Foxn1[+] TEC as early as E9.5 and E10.5 of gestation. In addition, K19 is also detectable within the parathyroid and thyroid, as well as endoderm of the 1st and 2nd pharyngeal pouches. Thus, K19 may be widely expressed within endodermal cells and multiple organ derivatives of the pharyngeal region and not simply limited to the TEC lineage.

As the timing of the peak frequency of K19[+] TEC coincides with the presence of organised cortex/medulla areas[22,23], we used confocal microscopy to determine their anatomical location in the developing thymus. Analysis of the mTEC marker, ERTR5, enabled us to identify cortex and medulla areas (Fig. 2a), and we found that K19[+] TEC were widely dispersed throughout these two separate compartments in thymus sections (Fig. 2a). Thus, K19[+] TEC were present in cortical areas (Fig. 2b) and in the subcapsular cortical zone (Fig. 2c), as well as in medullary areas where they represented a subset of ERTR5[+] cells (Fig. 2d). Taken together, we show that the embryonic thymus contains a K19[+] TEC subset, the frequency of which is developmentally

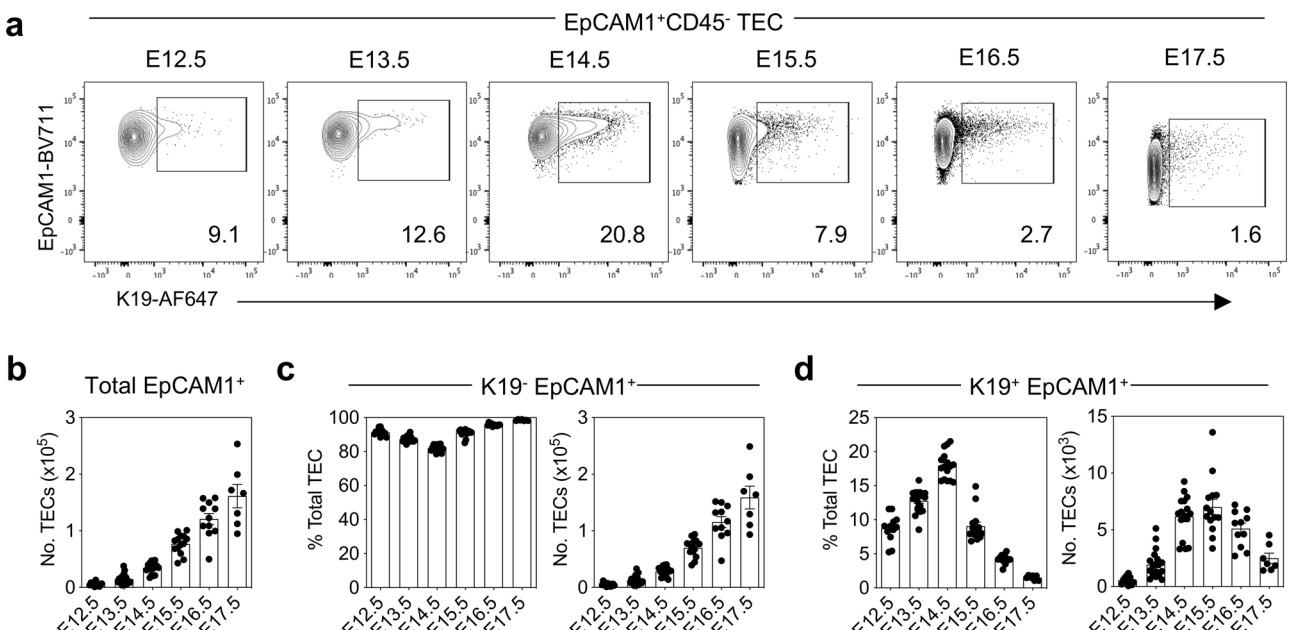

**Fig. 1 | K19 is expressed by a subset of embryonic TEC. a** Representative FACS plots showing expression of K19 by EpCAM1[+] cells during ontogeny. **b** Numbers of total EpCAM1[+] cells. **c** Numbers and proportions of K19[-] EpCAM1[+] cells. **d** Numbers and proportion of K19[+] EpCAM1[+] cells (E12.5 $n = 12$, E13.5 $n = 20$, E14.5 $n = 16$, E15.5 $n = 15$, E16.5 $n = 11$, E17.5 $n = 7$, from three independent experiments per gestational age). The data are shown as mean ± SEM.

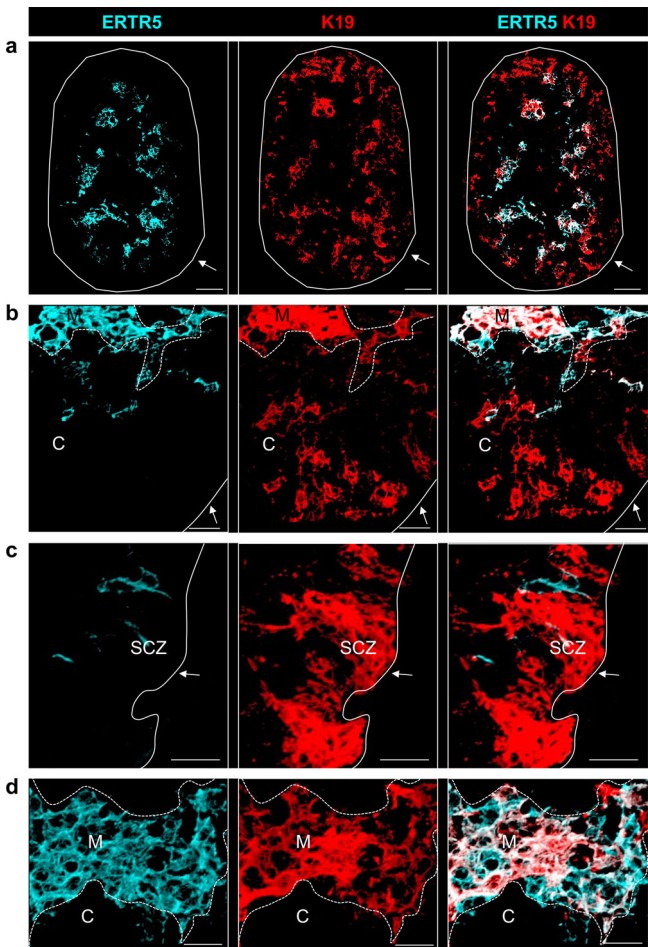

**Fig. 2 | K19⁺ TEC are widely distributed within thymic microenvironments.**
**a** Immunofluorescence of an E15.5 thymus lobe stained for ERTR5 (turquoise) and K19 (red), Scale bar, 50 μm. **b** K19 expression within ERTR5⁻ cortical regions. Scale bar, 20 μm. **c** K19 expression within the subcapsular zone. **d** K19 expression within ERTR5⁺ medullary areas. Scale bars in **b**–**d**, 20 m. SCZ subcapsular zone, C cortex, M medulla. Solid white lines indicate the edge of the thymus; dashed white lines indicate the boundary between cortex and medulla. Images are representative of 6 thymi, from 3 independent experiments.

regulated during thymus organogenesis, and which is dispersed throughout developing cortical and medullary areas.

## K19 expression defines the initial MHCII^neg stages of embryonic TEC development

To relate K19⁺ TEC to cTEC/mTEC lineages, we first defined embryonic TEC populations in the E12.5–E17.5 period using a panel of markers relevant to TEC development. Analysis of UEA1 and Ly51 within EpCAM1⁺ TEC showed that Ly51⁺UEA1⁻ TEC, likely to include previously described 'cTEC-like progenitors'[27–29], appeared prior to cells bearing the mTEC marker, UEA1, which appeared as a distinct population by E14.5 (S. Fig. 1A). Interestingly, quantification of Ly51⁺UEA1⁻ and Ly51⁻UEA1⁺ cells (S. Fig. 1B) indicated that neither population shared the same transient appearance as the K19⁺ TEC (Fig. 1). Moreover, within Ly51⁻UEA1⁺ cells, differences in levels of expression of MHCII and CD80 indicated a developmental sequence consisting of three discrete subsets. The first UEA1⁺ cells at E13.5 were MHCII^neg CD80^neg, which was followed by the appearance of MHCII^int CD80^neg then MHCII^hi CD80^hi cells at later gestational stages, suggesting progression from MHCII^neg to MHCII^int and then MHCII^hi (S. Fig. 1C). Similarly, the first Ly51⁺ cells at E12.5 were MHCII⁻, with MHCII⁺ cells emerging later, indicating an MHCII⁻ to MHCII⁺ progression for these cells (S. Fig. 1D). When we

analysed these markers in relation to K19 expression, we observed multiple Ly51/UEA1 subsets within E15.5 K19⁺ TEC (Fig. 3a). Compared to their K19⁻ counterparts, K19⁺ cells were enriched for an Ly51⁻UEA1⁺ phenotype (Fig. 3a). Importantly, K19⁺ cells were most abundant within the MHCII⁻ fraction of both Ly51⁻UEA1⁺ (Fig. 3b) and Ly51⁺UEA1⁻ cells (Fig. 3C). Moreover, the first Aire⁺ mTEC^hi present at E15.5 of gestation lacked K19 expression (Fig. 3b). Collectively, these findings define the phenotypic properties of K19⁺ TEC in relation to known cTEC/mTEC markers, and suggest that K19⁺ cells reside largely within immature fractions of Ly51⁺UEA1⁻ and Ly51⁻UEA1⁺ cells.

## K19⁺ TEC generate multiple functionally distinct mTEC subsets

To examine the developmental potential of embryonic K19⁺ TEC, we adopted an inducible fate-mapping approach using K19^CreERT[30,] and R26R^LSL tdTomato[31] strains to create Krt19^CreERT tdTomato^LSL mice (herein abbreviated as Krt19^CreERT TdTom). To induce K19^Cre expression in embryos, and fate-map cells within a fixed time window, we first gavaged pregnant females at E15.5 of gestation with a single dose of 4 mg tamoxifen. Using this approach, 24 h post gavage, 10% of TEC that stained positively with anti-K19 antibody were fate-mapped (S. Fig. 2A, B). When pups from tamoxifen-treated dams were harvested on the day of birth (PN d0) we found tdTom expression in thymus was limited to EpCAM1⁺ TEC, where 1% of total TEC were labelled (Fig. 4a). Expression of tdTom was absent from both EpCAM1⁻ non-TEC stroma and CD45⁺ haemopoietic cells (S. Fig. 2A). Importantly, flow cytometric analysis of digested neonatal thymus showed that tdTom⁺ cells were contained predominantly within the Ly51⁻UEA1⁺ subset (Fig. 4b), with the majority (72% ± 2.3%) of tdTom⁺ TEC cells having an Ly51⁻UEA1⁺ mTEC phenotype (Fig. 4c, d). In contrast, while Ly51⁺UEA1⁻ cells were readily detectable within non-fate-mapped TEC, tdTom labelling in these cells was barely detectable (5.9% ± 0.8%). Moreover, while non-fate-mapped Ly51⁺UEA1⁻ cells were uniformly MHCII⁺, ~50% of the few detectable fate-mapped Ly51⁺UEA1⁻ cells were MHCII⁻, a phenotype consistent with an immature stage of development (Fig. 4g). This bias towards mTEC generation resulted in a sharp skewing of the mTEC:cTEC ratio towards mTEC in fate-mapped cells (Fig. 4E). Moreover, confocal analysis showed that tdTom⁺ cells were present exclusively with medullary but not cortical thymic areas (Fig. 4f). These results indicate that fate-mapping of K19⁺ TEC on a single gestational day results in the presence of labelled cells at birth, which by both phenotype and intrathymic location are heavily biased towards the mTEC lineage. Indeed, the increase in frequency of tdTom⁺ cells 24 hours after Cre induction (~0.1%, S. Fig. 2) versus P0 (~1%, Fig. 4a) may be consistent with the expansion of K19⁺ progenitors and/or their downstream progeny.

As the mTEC compartment is known to contain multiple subsets that are functionally distinct, and given the importance of this heterogeneity for thymus medulla function, we further examined the mTEC derived from K19⁺ cells. Importantly, K19 fate-mapped tdTom⁺ mTEC were a mixture of MHCII^lo CD80^lo and MHCII^hi CD80^hi cells, and contained multiple subsets including Aire⁺ mTEC^hi, DCLK1⁺ tuft cells and CCL21⁺ mTEC (Fig. 4h–j). Importantly, this labelling of Aire⁺ mTEC and tuft cells is unlikely to explained by expression of K19 by these cells at the time of Cre recombination. First, we limited Cre induction to a 24-hour period from E15.5 of gestation and Aire⁺ mTEC present at this stage lack K19 expression (Fig. 3b). Second, tuft cells appear postnatally[9] and are therefore not available to undergo fate-mapping following Cre recombination in the embryo. Rather, our data suggest that fate-mapped tuft cells and Aire⁺ mTEC detectable at birth following Cre induction at E15.5 are both generated from immature progenitors.

Given the importance of sustained mTEC production for thymus medulla function, we examined whether embryonic K19⁺ TEC might support continued mTEC production into adulthood. In initial experiments, we attempted to study the thymus of adult mice born

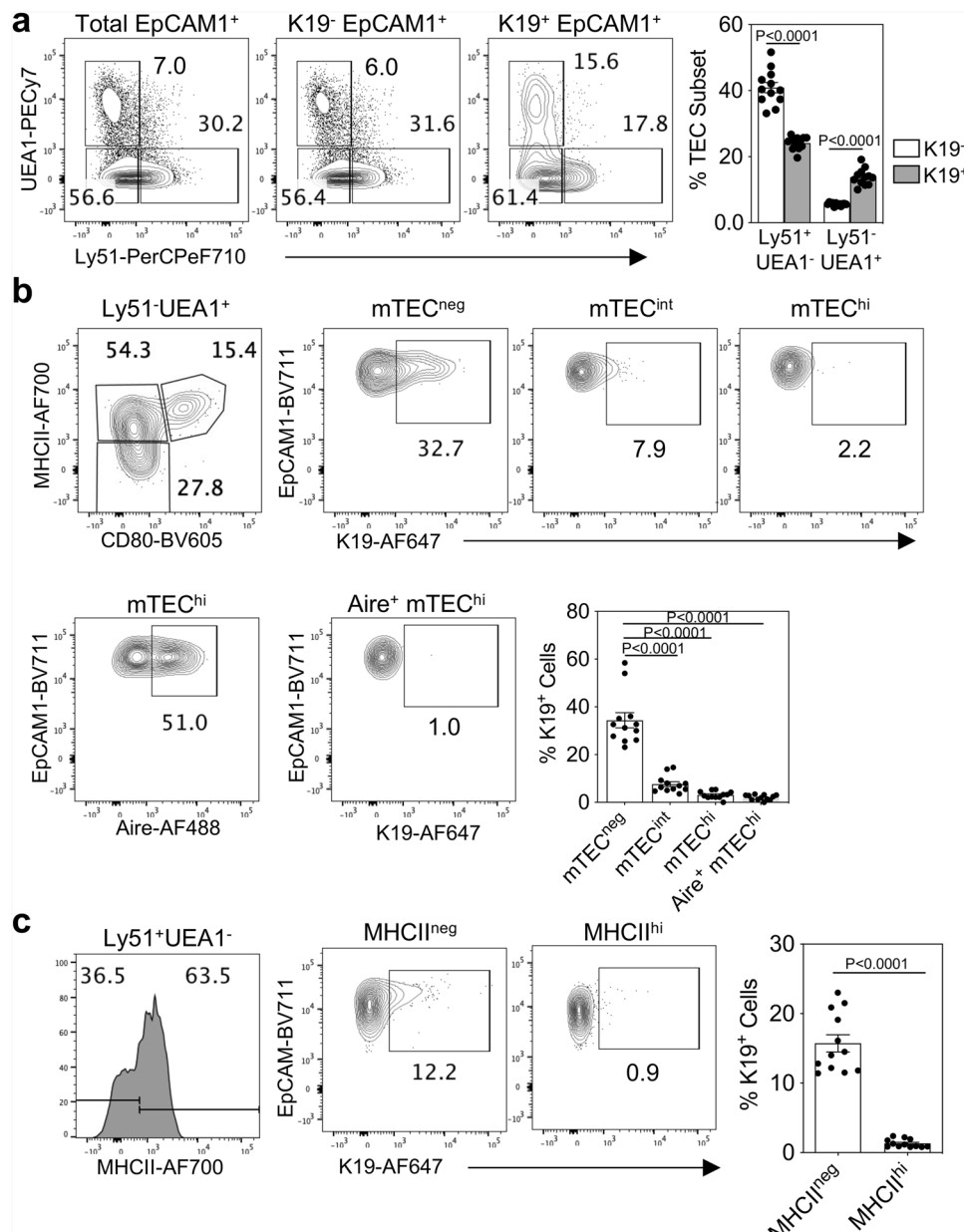

**Fig. 3 | K19 is specific to immature MHCII$^{neg}$ stages of TEC development.**
**a** Representative FACS plots showing expression of UEA1 and Ly51 by total EpCAM1$^+$ cells, K19$^-$EpCAM1$^+$ cells and K19$^+$EpCAM1$^+$ cells at E15.5, and corresponding quantitation ($n = 12$, from 3 independent experiments). Data analysed using a Student's t-test. **b** Representative FACS plots showing expression of MHCII and CD80 to define mTEC$^{neg}$ (MHCII$^-$CD80$^-$), mTEC$^{int}$ (MHCII$^{int}$CD80$^{int}$) and mTEC$^{hi}$ (MHCII$^{hi}$CD80$^{hi}$), and the corresponding expression of K19 by these subsets (upper panel), and expression of Aire within mTEC$^{hi}$, and the corresponding expression of K19 by Aire$^+$ mTEC$^{hi}$ (lower panel). Bar chart shows proportion of K19$^+$ cells within mTEC$^{neg,}$ mTEC$^{int,}$ and mTEC$^{hi}$ at E15.5 ($n = 12$, from 3 independent experiments). Data analysed using a one-way ANOVA, with Bonferroni post hoc test. **c** Representative FACS plots showing expression of MHCII within UEA1$^-$Ly51$^+$ cells, and expression of K19 by these subpopulations. Bar chart shows proportion of K19$^+$ cells within MHCII$^-$UEA1$^-$Ly51$^+$ and MHCII$^+$UEA1$^-$Ly51$^+$ TEC at E15.5 ($n = 12$, from 3 independent experiments). Data analysed using a Student's t test. The data are shown as mean ± SEM.

from pregnant mothers that received tamoxifen at E15.5 of gestation. However, as previously reported[32], tamoxifen treatment during embryogenesis caused significant lethality during postnatal stages, which prevented use of this approach to analyse long term progeny analysis of embryonic K19$^+$ cells. As an alternative, and to avoid such limitations, we adopted a thymus transplant approach. Here, Cre recombinase activity was induced within *Krt19*$^{CreERT}$TdTom embryos by administration of a single tamoxifen dose into pregnant female mice at E15.5 of pregnancy. Thymus lobes from embryos were harvested 24 h later and grafted under the kidney capsule of adult WT mice (Fig. 5a). Importantly, at day 31 post-transplant, the equivalent of 4 weeks of

postnatal age for the transplanted thymus, we saw the continued presence of tdTom$^+$ cells (~0.5% of total TEC), the vast majority of which were of an Ly51$^-$UEA1$^+$ mTEC phenotype (Fig. 5b, c) that contained multiple mTEC subsets within them, including Aire$^+$ mTEC$^{hi}$, tuft cells and CCL21$^+$ mTEC (Fig. 5d) within medullary areas (Fig. 5e). Moreover, when fate-mapped embryonic thymus grafts were harvested at a timepoint equivalent to 8 weeks of postnatal age, TdTom$^+$ TEC continued to be detected (Fig. 5f), which were heavily biased towards Ly51$^+$UEA1$^-$ mTEC and consisted of mTEC$^{lo}$, mTEC$^{hi}$, Aire$^+$ mTEC, CCL21$^+$ mTEC and tuft cells (Fig. 5h, i). Of note, in fate-mapping experiments harvested at P0 compared to those harvested at the

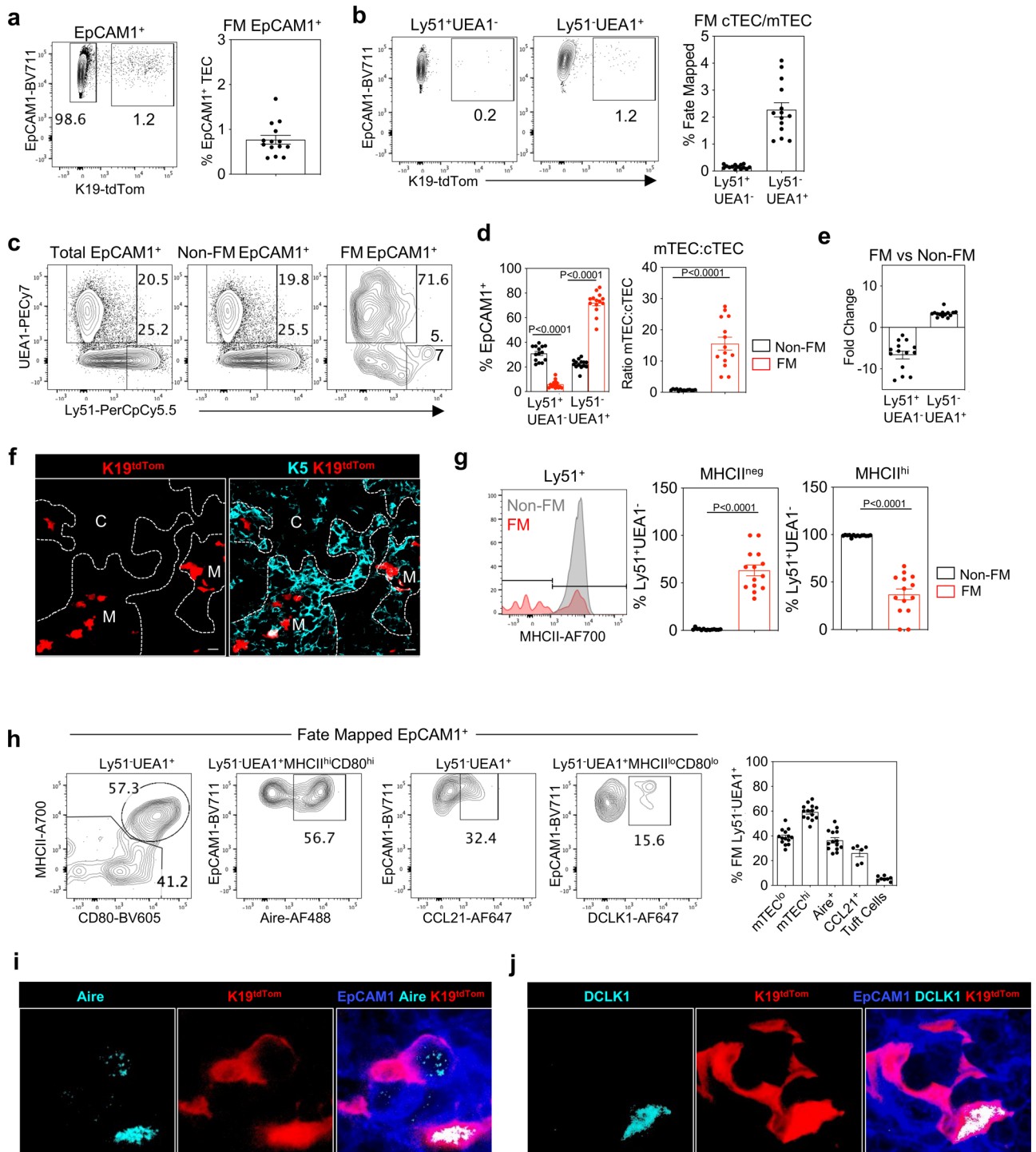

**Fig. 4 | K19 identifies embryonic multipotent mTEC progenitors (mmTECp). a** K19[Cre] was induced in *Krt19*[CreERT]tdTom embryos at E15.5 via the administration of tamoxifen to pregnant mice, and fate-mapped thymi were harvested at PNd0 (*n* = 14, from 3 independent experiments). Fate-mapped cells were detected within EpCAM1[+] cells by flow cytometry and quantitated. **b** Representative FACS plots and quantitation of K19-tdTom fate-mapped cells within EpCAM1[+]UEA1[+] and EpCAM1[+]Ly51[+] TEC. **c** Expression of Ly51 and UEA1 by total EpCAM1[+] cells, K19-tdTom[-] EpCAM1[+] cells (non-fate-mapped) and K19-tdTom[+]EpCAM1[+] cells (fate-mapped). **d** Bar charts show proportions of Ly51[+]UEA1[-] and Ly51[-]UEA1[+] cells, and mTEC:cTEC ratio within non-fate-mapped and fate-mapped TEC. **e** Bar chart shows fold change in Ly51[+]UEA1[-] and Ly51[-]UEA1[+] cells within non-fate-mapped and fate-mapped TEC. **f** Immunofluorescence of PNd0 thymi following K19-fate-mapping at

E15.5. K19-tdTom (red) and K5 (turquoise). 'C', and 'M' indicate cortex and medullary areas respectively. Scale bar denotes 20 µm. Image representative of 4 thymi. **g** Expression of MHCII by EpCAM1[+]Ly51[+] fate-mapped (FM) or non-fate-mapped (Non-FM) TEC. **h** Representative FACS plots illustrating the phenotype of fate-mapped mTEC subsets. mTEC[hi] (MHCII[hi]CD80[hi], *n* = 14), mTEC[lo] (MHCII[lo]CD80[lo], *n* = 14), Aire[+] (MHCII[hi]CD80[hi]Aire[+], *n* = 14), CCL21[+] (*n* = 6), tuft cells (MHCII[lo]CD80[lo]DCLK1[+], *n* = 8), and corresponding quantitation. **i, j** Immunofluorescence of PNd0 thymi where K19[Cre] was induced at E15.5. Scale bar denotes 10 µm. Image representative of 4 thymi, from 3 independent experiments. K19-tdTom (red), EpCAM1 (blue), **f** Aire (turquoise), **g** DCLK1 (turquoise). Data analysed using a Student's *t* test. The data are shown as mean ± SEM.

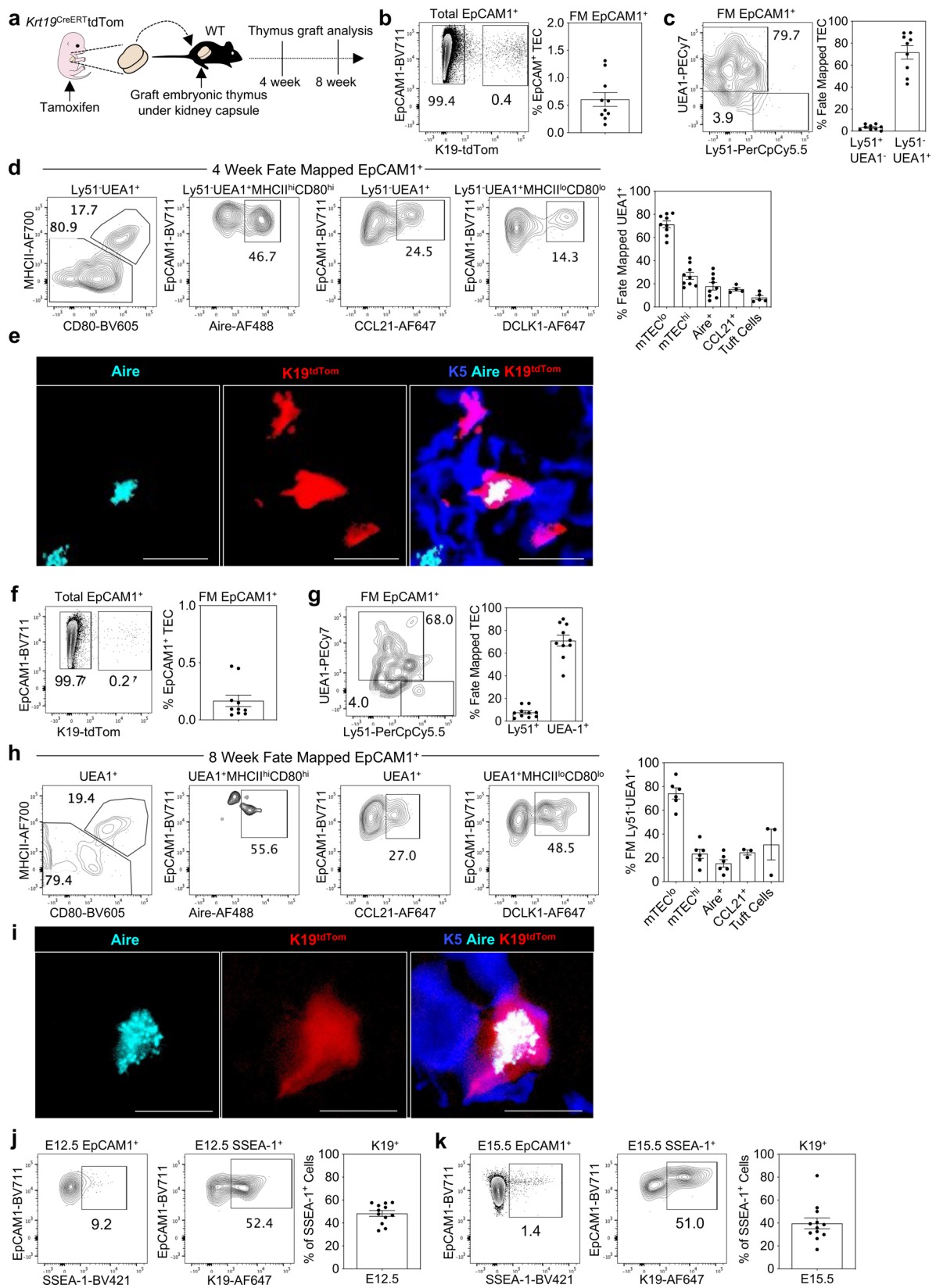

equivalent of both 4 and 8 weeks of age, we saw differences in the frequencies of fate-mapped mTEC$^{lo}$ and mTEC$^{hi}$ cells. Here, fate-mapped mTEC$^{hi}$ dominated at P0 (Fig. 4h) whilst mTEC$^{lo}$ dominated at 4 (Fig. 5d) and 8 weeks (Fig. 5h). It is not clear whether these changes reflect unequal expansion of fate-mapped cells within individual subsets, the turnover of Aire$^+$ cells within mTEC$^{hi}$, or the progressive

accumulation of post-Aire mTEC$^{lo}$ stages. Taken together, these findings demonstrate that embryonic K19$^+$ TEC are capable of the long-term generation of multiple mTEC subsets, providing evidence that an embryonic mTEC progenitor sustains mTEC diversity in adulthood.

In previous studies, SSEA1 expression was shown to identify a subset of embryonic TEC with stem cell properties, including self-

**Fig. 5 | Sustained generation of mTEC diversity from embryonic K19+ mmTECp.**
**a** K19Cre was induced in *Krt19*CreERTtdTom embryos at E15.5 and after 24 h thymi were grafted under the kidney capsule of WT mice. Thymus grafts were harvested at the equivalent of postnatal week 4 and 8. **b** At 4 weeks, fate-mapped cells were detected within EpCAM1+ cells by flow cytometry and quantitated (*n* = 9, from 4 independent experiments). **c** Representative FACS plots and quantitation of K19-tdTom fate-mapped cells at 4 weeks within EpCAM1+UEA1+ and EpCAM1+Ly51+ TEC.
**d** Representative FACS plots illustrating the phenotype of fate-mapped mTEC subsets within fate-mapped thymus grafts at postnatal week 4. mTEChi (MHCII-hiCD80hi, *n* = 9), mTEClo (MHCIIloCD80lo, *n* = 9), Aire+ (MHCIIhiCD80hiAire+, *n* = 9), CCL21+ (*n* = 4) and tuft cells (MHCIIloCD80loDCLK1+, *n* = 5), and corresponding quantitation. **e** Immunofluorescence of fate-mapped thymi at postnatal week 4, Aire (turquoise) K19-tdTom (red), K5 (blue). Scale bar denotes 10 μm. Image representative of 3 grafts. **f** At 8 weeks, fate-mapped cells were detected within

EpCAM1+ cells by flow cytometry and quantitated (*n* = 10, from 4 independent experiments). **g** Representative FACS plots and quantitation of K19-tdTom fate-mapped cells at 8 weeks within EpCAM1+UEA1+ and EpCAM1+Ly51+ TEC.
**h** Representative FACS plots illustrating the phenotype of fate-mapped mTEC subsets within fate-mapped thymus grafts at postnatal week 8. mTEChi (MHCII-hiCD80hi, *n* = 8), mTEClo (MHCIIloCD80lo, *n* = 8), Aire+ (MHCIIhiCD80hiAire+, *n* = 8), CCL21+ (*n* = 5) and tuft cells (MHCIIloCD80loDCLK1+, *n* = 4), and corresponding quantitation. **i** Immunofluorescence of fate-mapped thymi at postnatal week 8, Aire (turquoise) K19-tdTom (red), K5 (blue). Scale bar, 10 μm. Image representative of 3 grafts, from 3 independent experiments. **j** Representative FACS plots of SSEA1 and K19 expression at E12.5 (**j**) and E15.5 (**k**), and quantitation of K19+ SSEA-1+ TEC (*n* = 12 for both stages, from 3 independent experiments). The data are shown as mean ± SEM.

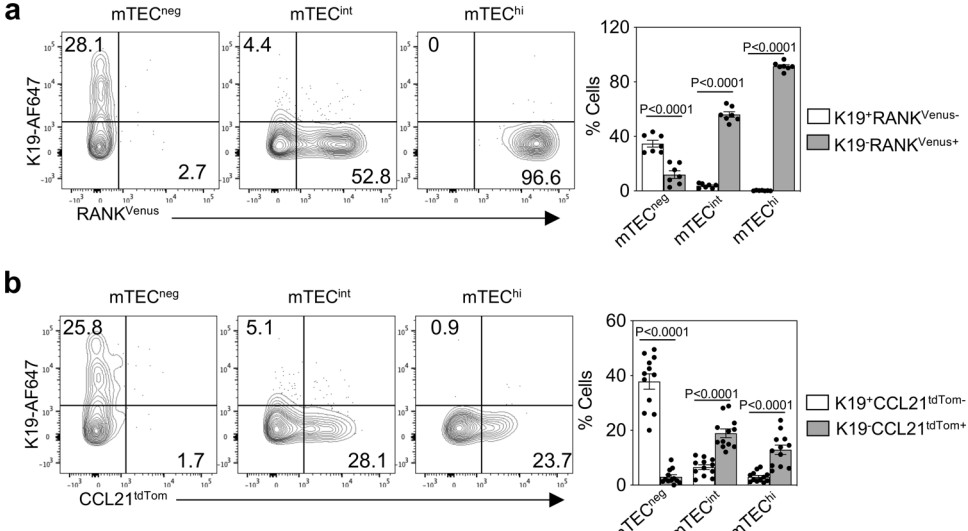

**Fig. 6 | K19+ TEC appear prior to RANK and CCL21 stages of mTEC development.**
**a** Representative FACS plots showing expression of K19 and RANKVenus by mTECneg (MHCII-CD80-), mTECint (MHCIIintCD80int) and mTEChi (MHCIIhiCD80hi) at E15.5, and corresponding quantitation, *n* = 7, from 3 independent experiments.

**b** Representative FACS plots showing expression of K19 and CCL21tdTom by mTECneg (MHCII-CD80-), mTECint (MHCIIintCD80int) and mTEChi (MHCIIhiCD80hi) at E15.5, and corresponding quantitation, *n* = 12, from 3 independent experiments. Data analysed using a Student's *t* test. The data are shown as mean ± SEM.

renewal potential and the ability to support the long term generation of Aire+ mTEC[21,33,34]. Here, it is important to note that our study focuses on identifying mTEC progenitors and investigating their potential developmental in relation to multiple functionally distinct mTEC subsets, rather than examining their possible stem cell self-renewal properties. However, in an initial attempt to relate K19+ mTEC progenitors to mTEC stem cells, we analysed E12.5 (Fig. 5j) and E15.5 (Fig. 5k) TEC for expression of K19 alongside SSEA1, the marker used by Sekai et al. to identify mTEC stem cells[21]. Interestingly, at both developmental stages, SSEA1+ cells were found to be heterogeneous for K19 expression, with ~40–50% of SSEA1+ cells expressing K19 (Fig. 5j, k). Whether this heterogeneity relates to differences in developmental potential and/or self-renewal within embryonic SSEA1+ TEC requires further examination in future studies.

**K19+ progenitors lack mTEC lineage hallmarks at multiple developmental stages**
One explanation for the ability of embryonic K19+ TEC to give rise to multiple mTEC subsets is that at E15.5 of gestation, they are a heterogeneous mixture of progenitors and mature mTEC. To investigate this further, we adopted two approaches. First, we examined E15.5 K19+ TEC for their expression of known hallmarks of mTEC maturity using RANKVenus [35] and CCL21tdTom [3] reporter mice and flow cytometry using

anti-K19 antibodies. Consistent with findings described above (Fig. 3), K19+ TEC at E15.5 were contained within the EpCAM1+UEA1+Ly51-MHCII-CD80- mTECneg fraction, and absent from mTECint and mTEChi fractions (Fig. 6a). Importantly, we found that K19+ TEC lacked expression of the mTEC markers RANKVenus (Fig. 6a) and CCL21tdTom (Fig. 6b), with both markers being readily detectable in both mTECint and mTEChi subsets. Thus, E15.5 K19+ TEC that generate multiple mTEC subsets represent a stage in the mTEC lineage that occurs prior to expression of RANK and CCL21. As a second approach, we performed fate-mapping experiments with tamoxifen administration at E12.5 of gestation. Importantly, UEA1+ mTEC are absent at this timepoint, yet K19+ cells are still detectable within Ly51+UEA1- cells (Fig. 7a). Fate-mapping was induced in *Krt19*CreERTTdTom embryos at E12.5, and on the day of birth, neonatal thymi were harvested and analysed by confocal microscopy and flow cytometry. Again, tdTom+ cells (~2% of total TEC, Fig. 7b) were detectable exclusively within medullary areas (Fig. 7c), and showed a strong bias in their generation of mTEC (Fig. 7d), which included Aire+ mTEChi, tuft cells and CCL21+ mTEC (Fig. 7e, f). Thus, embryonically K19-fate-mapped cells at both E12.5 and E15.5 of gestation give rise to multiple functionally distinct mTEC subsets, indicating that K19+ cells in this developmental period are multipotent mTEC progenitors (mmTECp) that support the formation of epithelial diversity in the thymus medulla.

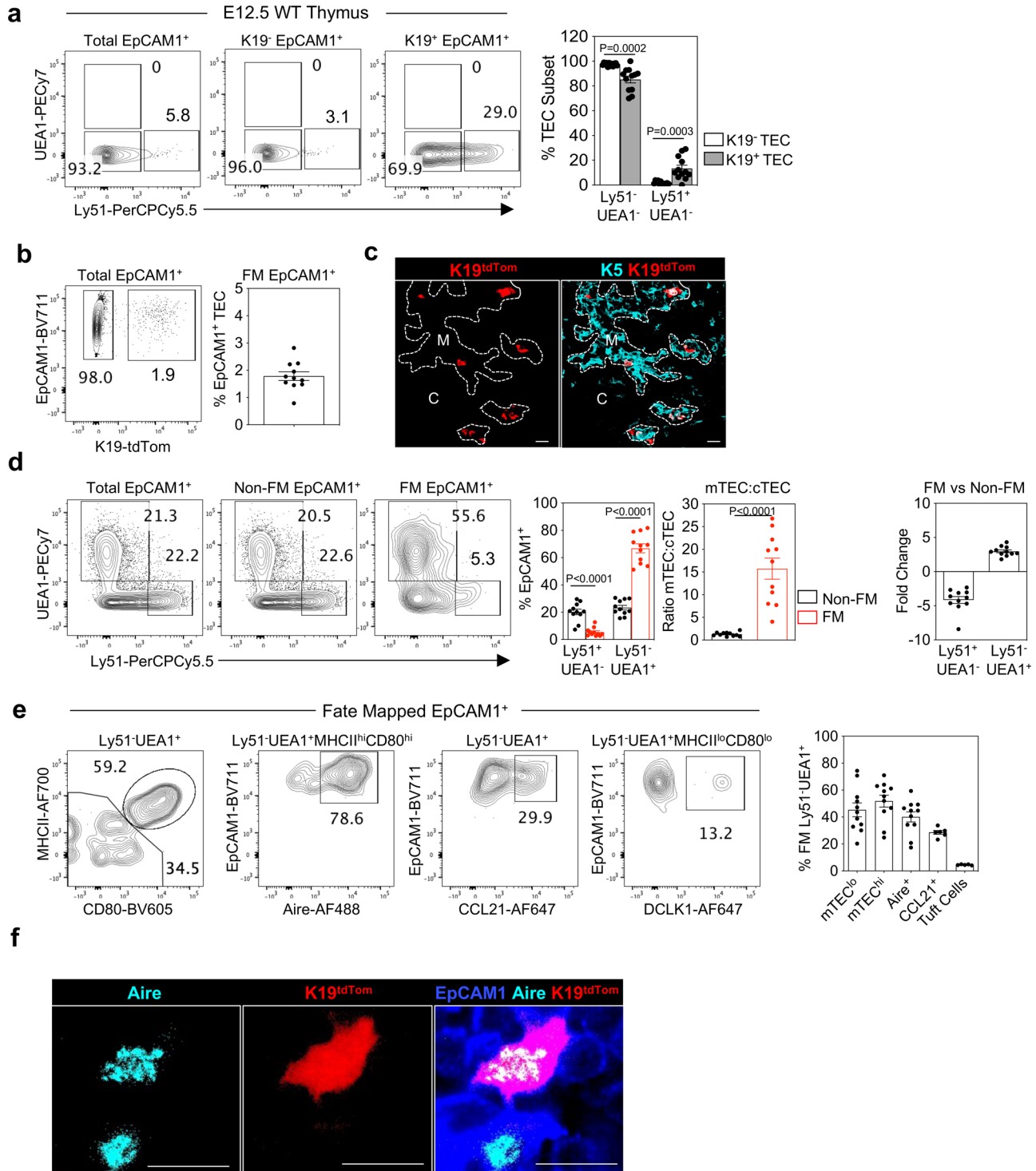

**Fig. 7 | K19+ mmTECp are present at E12.5 of development. a** Representative FACS plots showing expression of UEA1 and Ly51 by total EpCAM1+ cells, K19- EpCAM1+ cells and K19+EpCAM1+ cells at E12.5, and corresponding quantitation, $n = 12$, from 3 independent experiments. **b** $Krt19^{CreERT}$tdTom embryos were fate-mapped via tamoxifen administration at E12.5, and the neonatal thymus was harvested at birth. Fate-mapped cells were quantitated ($n = 11$, from 3 independent experiments) **c** Immunofluorescence of fate-mapped thymi at PNd0, K5 (turquoise) K19-tdTom (red), Scale bar denotes 20um. Image representative of 4 thymi, from 3 independent experiments. **d** Expression of Ly51 and UEA1 by total EpCAM1+ cells, K19-Tom- EpCAM1+ cells (non-fate-mapped), and K19-tdTom+EpCAM1+ cells (fate-mapped) at PNd0, bar charts show corresponding quantitation, $n = 11$, from 3 independent experiments. **e** Representative FACS plots illustrating the phenotype of fate-mapped mTEC subsets within PNd0 fate-mapped thymus. mTEC^hi (MHCII^hiCD80^hi, $n = 11$), mTEC^lo (MHCII^loCD80^lo, $n = 11$), Aire+ (MHCII^hiCD80^hiAire+, $n = 11$), CCL21+ ($n = 6$) and tuft cells (MHCII^loCD80^loDCLK1+, $n = 5$) and corresponding quantitation. **f** Immunofluorescence of fate-mapped thymi at PNd0, Aire (turquoise) K19-tdTom (red), EpCAM1 (blue). Scale bar,10 μm. Image representative of 4 thymi. Data analysed using a Student's $t$ test. The data are shown as mean ± SEM.

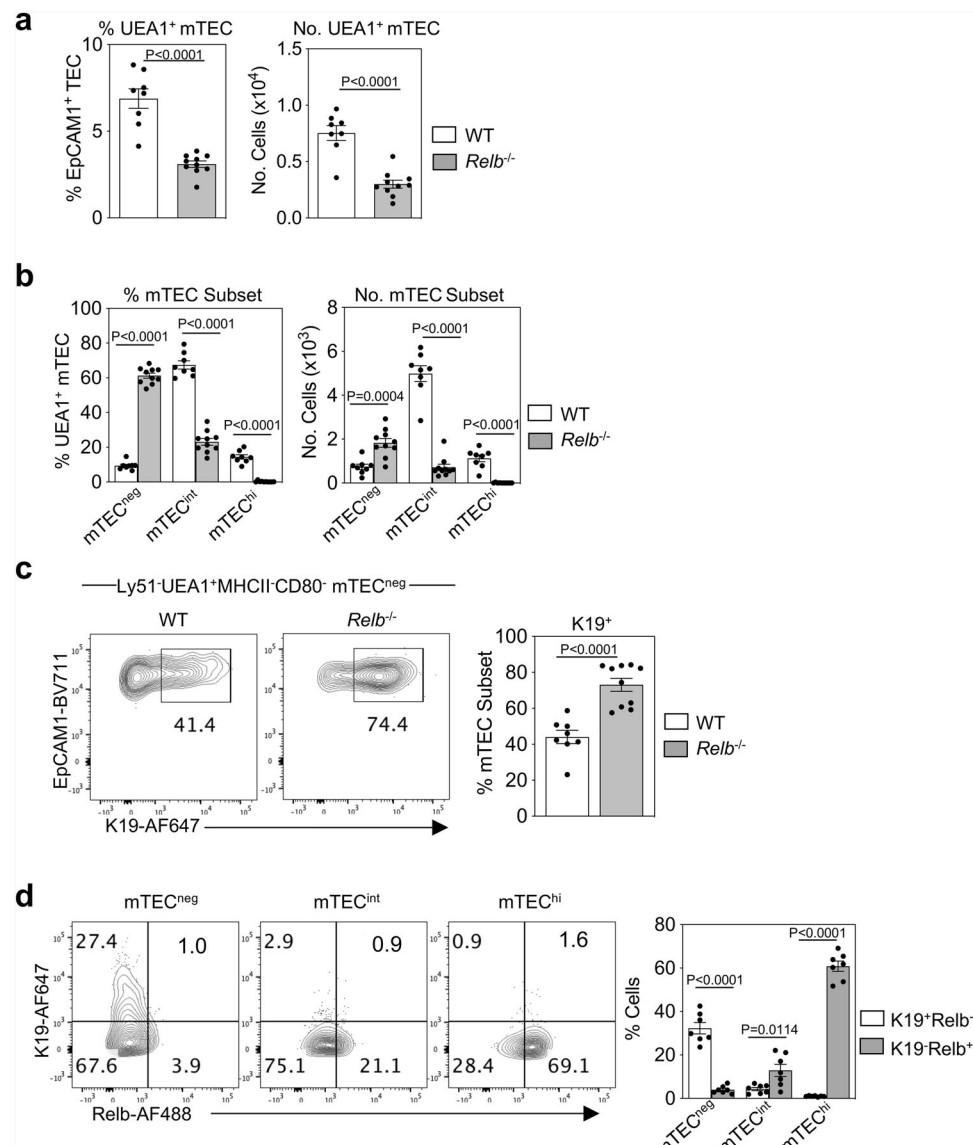

**Fig. 8 | K19⁺ mmTECp are upstream and independent of Relb-dependent mTEC stages. a** Proportions and numbers of Ly51⁻UEA1⁺ mTEC in E15.5 WT and Relb⁻/⁻ thymus. **b** Proportions and numbers of MHCII⁻CD80⁻ mTECⁿᵉᵍ, MHCIIⁱⁿᵗCD80⁻ mTECⁱⁿᵗ, MHCIIʰⁱCD80ʰⁱ mTECʰⁱ in E15.5 WT and Relb⁻/⁻ thymus. **c** Representative FACS plots and corresponding quantitation showing K19 expression within Ly51⁻ UEA1⁺MHCII⁻CD80⁻mTECⁿᵉᵍ in WT and Relb⁻/⁻ thymi at E15.5. **a–c** WT *n* = 8, Relb⁻/⁻ *n* = 11, from 3 independent experiments. **d** Representative FACS plots showing expression of K19 and Relb by mTECⁿᵉᵍ, mTECⁱⁿᵗ and mTECʰⁱ in WT E15.5 thymi, and corresponding quantitation, *n* = 7, from 3 independent experiments. Data analysed using a Student's *t* test. The data are shown as mean ± SEM.

## mmTECp arise independently of the mTEC regulator Relb

Mice lacking the NF-κB signalling component, Relb, display an early block in mTEC lineage development that results in the absence of mature mTEC and a failure to form organised medullary areas[36,37]. However, the patterns of Relb expression in mTEC and mTEC progenitors are poorly understood. Given that K19⁺ mmTECp represent an early stage in mTEC lineage development, we examined the presence of these cells in E15.5 *Relb⁻/⁻* mice, and evaluated Relb expression during early stages of mTEC development in WT mice, including K19⁺mmTECp. Consistent with an early block in mTEC development in *Relb⁻/⁻* embryos, flow cytometric analysis of digested E15.5 thymus lobes showed a marked reduction in UEA1⁺ mTEC compared with WT controls (Fig. 8a). Interestingly, unlike WT controls, the majority of the remaining Ly51⁻UEA1⁺ TEC in *Relb⁻/⁻* embryos were of an immature mTECⁿᵉᵍ phenotype (Fig. 8b). Importantly, K19 expression was detectable in mTECⁿᵉᵍ in *Relb⁻/⁻* mice (Fig. 8c). Indeed, a greater proportion of mTECⁿᵉᵍ cells expressed K19 in *Relb⁻/⁻* mice compared to WT

(Fig. 8c), suggesting that Relb is not essential for the development of K19⁺ mmTECp, but is required for their transition to the K19⁻ mTECⁱⁿᵗ stage. To examine this further, we performed simultaneous flow cytometric analysis of K19 and Relb expression on E15.5 mTEC subsets present in WT mice. While Relb was readily detectable in mTECⁱⁿᵗ and mTECʰⁱ stages, K19⁺ mmTECp lacked expression of Relb (Fig. 8d). Thus, K19⁺ mmTECp appear prior to expression of Relb in the mTEC lineage. This provides an explanation for their presence in *Relb⁻/⁻* mice, and defines the timing of requirement for Relb expression during mTEC development.

## CD9 expression defines mmTECp in WT mice

The findings described above identify K19 as a hallmark of mmTECp through use of anti-K19 antibodies. This approach requires intracellular staining and cell permeabilisation which then limits the isolation of viable mmTECp for functional studies. To aid in the identification and further study of K19⁺ mmTECp in WT mice, we performed

massively parallel flow cytometry of E15.5 TEC for 259 exploratory cell surface markers. With this approach, we aimed to learn more about the phenotypic properties of K19+ TEC, with a view to aiding future studies aimed at analysing and isolating TEC progenitor in WT mice. Here, to help limit the loss of cell surface proteins, we employed liberase for enzymatic digestion of thymus lobes. The information of the expression of 10 backbone markers, including K19, was used to compute the co-expression of all exploratory markers across the acquired cells by the machine learning algorithm Infinity Flow[38]. The resulting expression information was further analysed by the single-cell analysis pipeline Seurat, as recently described for postnatal TEC subsets[39]. Hierarchical clustering of the computed data resulted in 12 clusters, as illustrated by Uniform Manifold Approximation and Projection (Fig. 9a). Given that TEC populations in the E15.5 embryonic thymus are largely immature, and at least some TEC subsets present in adult thymus e.g. thymic tuft cells are absent, we simply labelled clusters numerically rather than attempt to assign clusters to cTEC and mTEC lineages. Differential expression analysis of the surface markers revealed K19 expression in clusters 9 and 10, being highest in the former (Fig. 9b, c). A consistent surface expression profile overlap with K19 was observed for CD9 (Fig. 9b, c). Indeed, highest expression of CD9 was also detected within clusters 9 and 10. Interestingly, some expression of UEA1 and Ly51 was evident within clusters 9 and 10, while MHCII was absent from cluster 9 (Fig. 9d). We confirmed expression of K19 by a mean value of $82.5 \pm 3.8\%$ of EpCAM1+MHCII-CD9hi E15.5 TEC (Fig. 9e). Importantly, while such initial findings provide a detailed phenotypic profile of K19+ progenitors and indicate overlap between CD9 and K19 expression, it currently remains unclear whether cells expressing CD9 have developmental properties similar to K19+ TEC progenitors. Further studies are required to investigate the developmental potential of CD9+ cells within MHCII- TEC, to determine whether CD9 may be a useful marker to help identify K19+ mmTECp in wild-type embryonic thymus.

## Discussion

In the thymus, epithelial microenvironments control the development and selection of immature thymocytes, which results in the generation of self-tolerant T cells that are essential for adaptive immune responses. In recent years, several advances have been made in revealing TEC heterogeneity. For example, recent studies have now identified multiple mature medullary thymic epithelial cells (mTEC) subsets that are transcriptionally and functionally distinct, enabling the medulla to play multiple non-overlapping roles in T-cell development[40]. In addition, progress is being made on understanding mTEC lineage specification, including recently described roles for Notch-Notch ligand interactions in early mTEC development[41–43]. Despite this, the nature of the mTEC progenitors that ensure continued generation and maintenance of mTEC diversity into adulthood are poorly understood. To address this, we searched for TEC progenitors during thymus ontogeny, and examined the ability of defined populations to sustain the long term production of multiple mTEC subsets. From as early as E12.5 of gestation, we identified a TEC subset expressing K19 that was present throughout development with a peak frequency around E15.5 of gestation. We found that K19+ TEC were present within multiple anatomically distinct regions of the embryonic thymus, including both cortical and medullary areas that first emerge during development. Importantly, fate-mapping of embryonic K19+ cells demonstrated their ability to generate multiple mTEC subsets that were detected at the equivalent of 8 weeks of adult life, including Aire+ and CCL21+ subsets, and DCLK1+ thymic tuft cells, demonstrating a common origin for mTEC functional diversity in the adult thymus. Interestingly, some K19+ cells in the embryonic thymus were detected in cortical areas and expressed an Ly51+UEA1- phenotype typical of the cTEC lineage. Moreover, while the large majority of K19 fate-mapped cells were of an Ly51-UEA1+ mTEC phenotype, a small percentage, around 5%, were

Ly51+UEA1-. Several possibilities may explain this observation. First, although K19+ embryonic TEC predominantly give rise to mTEC, they may also retain residual cTEC potential that remains from an upstream cTEC/mTEC bipotent progenitor stage. In this scenario, K19 may not mark cells fully committed to the mTEC lineage but identify cells that are mTEC-biased. Alternatively, Ly51+UEA1- fate-mapped cells may not reflect fully mature cTEC but instead are cTEC-like cells that express cTEC lineage markers but have been shown to possess mTEC potential[44]. That Ly51+ fate-mapped TEC contain both MHCII- and MHCII+ subsets may indeed provide support for both scenarios, where MHCII+Ly51+ fate-mapped cells represent residual cTEC potential, and MHCII-Ly51+ fate-mapped cells represent immature cTEC-like cells. Further studies that aid in distinguishing functionally mature cTEC from immature cTEC-like progenitors will aid in studying the cTEC/mTEC lineage divergence that takes place during TEC development.

The capacity of K19+ embryonic TEC to act as a common progenitor source of mature mTEC subsets was evident in their ability to support diversity within the first mTEC cohorts in the neonatal thymus. Moreover, embryonic K19+ cells also supported the long-term maintenance of mTEC diversity into adulthood, as indicated by the detection of diverse progeny up to a stage the equivalent of 8 weeks of postnatal age. Collectively, our findings identify a subset of embryonic TEC that is defined by K19 expression, and which serves as a multipotent mTEC progenitor (mmTECp) population that sustains mTEC diversity in adult life. Here, it is important to point out that the inducible fate-mapping approach used in this study, which limits Cre induction to a 24-h period during embryogenesis, results in the labelling of a proportion (~10%) of K19+ TEC. For this reason, we do not make conclusions regarding the relative contribution of K19+ TEC and K19- TEC to generation of the total mTEC compartment. Rather, we limit the interpretation of our data to the analysis and fate of those cells that have successfully undergone fate-mapping. Importantly, this allows us to make the conclusion that K19+ embryonic TEC are able to give rise to multiple functionally distinct mTEC subsets.

Our definition and analysis of mmTECp is important in relation to other studies examining the developmental origins of TEC populations. For example, previous studies described a population of embryonic TEC termed 'cTEC-like cells', that expressed cTEC markers yet showed evidence for mTEC potential, including their ability to give rise to Aire+ mTEC[27,28]. Interestingly, we found that some embryonic K19+ mmTECp expressed the cTEC marker Ly51, and resided within cortical areas, suggesting that cTEC-like cells with mTEC potential may be likely to be contained within the K19+ mmTECp fraction. It is also of interest that K19+ mmTECp lack expression of RANK, which makes them distinct from previously described RANK+ progenitors that have been described in the embryonic thymus[16,17]. Indeed, the presence of K19+ mmTECp at E12.5 of gestation suggests they represent a stage in mTEC development that is upstream of RANK+ progenitors. Here, it is also important to note that while RANK+ progenitors are known to give rise to Aire+ mTEC[45] their ability to give rise to other mTEC subsets, including CCL21+ mTEC and thymic tuft cells, has not been addressed. In contrast, we show that K19+ mmTECp give rise to Aire+ mTEC alongside CCL21+ mTEC and tuft cells. Thus, K19+ mmTECp represent an early stage in mTEC development that is upstream of RANK expression, with the subsequent acquisition of RANK by downstream progeny enabling Aire+ mTEC development. Consistent with this is the presence of K19+ mmTECp in Relb-deficient mice, where mTEC development is blocked at an early stage that is upstream of RANK expression[17]. How K19+ mmTECp described here relate to SSEA1+ mTEC stem cells[21] that were shown to contain cells with self-renewal potential, and also be capable of the long-term mTEC generation, is not clear. Here, it is important to note that the major goal of our study was to examine the developmental pathways that give rise to multiple functionally important mTEC subsets. As such, we focused on progenitor potential rather than stem cell characteristics. From this, a key finding

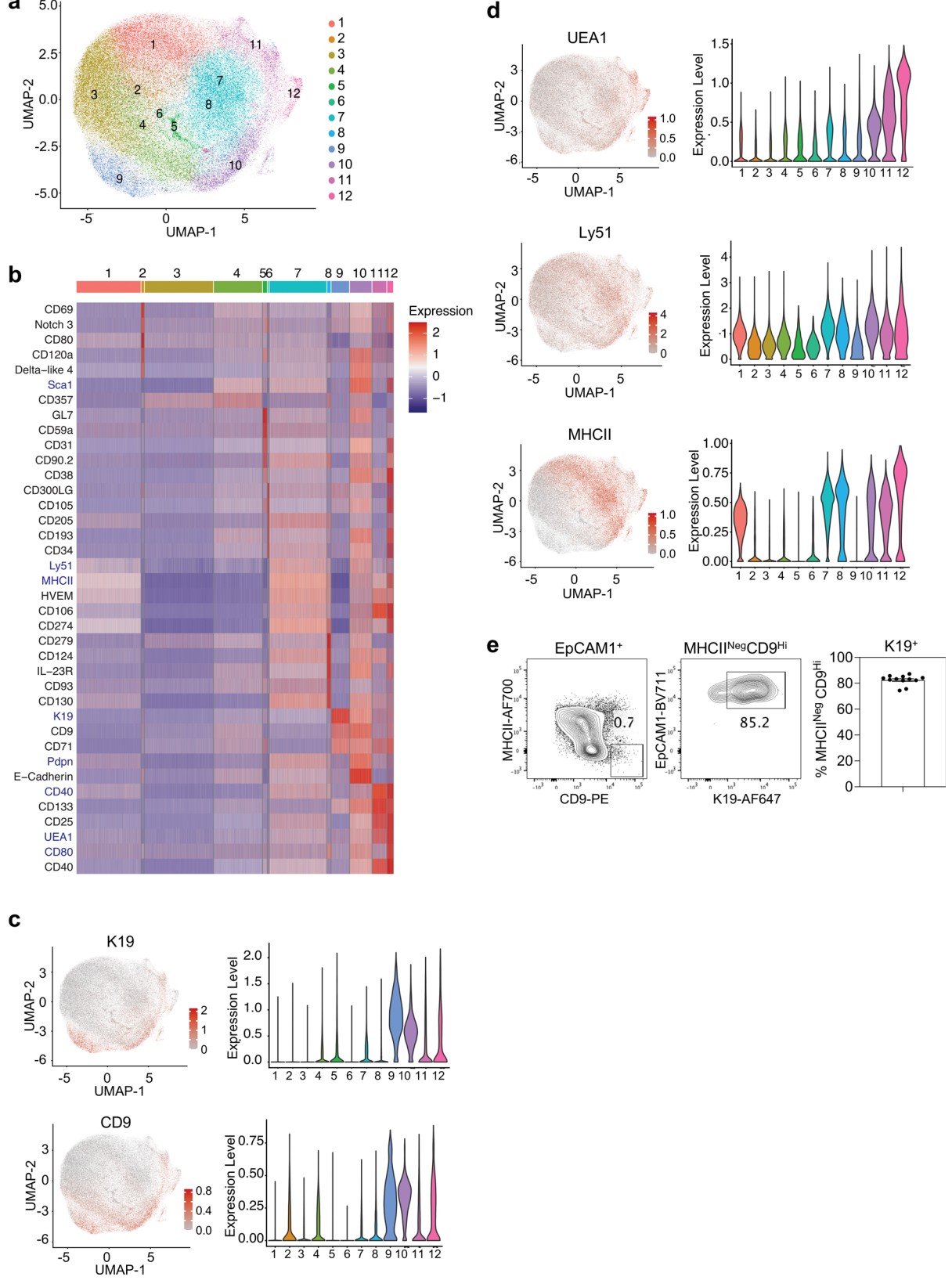

**Fig. 9 | CD9 expression defines mmTECp in wild-type mice. a–c** Infinity Flow analysis was used to impute the expression of surface markers on TEC (CD45⁻ EpCAM1⁺) derived from thymi of E15.5 WT embryos. **a** Hierarchical clustering analysis was performed on 121832 TEC and projected in a two-dimensional space using UMAP. Each colour represents a specific cluster as indicated. **b** Heatmap displays the expression of the top 5 markers upregulated in each cluster (log fold-change >0.15). Backbone markers have a blue font. **c** UMAP graphs and violin plots illustrating the expression of K19 and CD9. **d** UMAP graphs and violin plots illustrating the expression of UEA1, Ly51 and MHCII. Colour gradient indicates expression levels in the UMAP graphs and colours in the violin plots represent the different clusters, as defined in **a**. **e** Representative FACS plots showing expression of K19 by MHCII^lo^CD9^hi^ TEC and corresponding quantitation, $n = 12$, from 3 independent experiments. The data are shown as mean ± SEM.

of our study is that a K19[+] subset of embryonic TEC can give rise to Aire[+] cells, CCL21[+] cells and tuft cells, suggesting a common origin for mTEC diversity. While the possible self-renewal properties of K19[+] progenitors require further investigation, it is perhaps interesting that at both E12.5 and E15.5 of gestation, we found that approximately 40-45% of SSEA1[++] cells expressed K19. While such findings suggest potential heterogeneity within the SSEA1[+] embryonic TEC fraction previously shown to possess stem cell properties, further experiments are required to fully examine how K19 and SSEA1 expression relates to mTEC stem and/or progenitor properties.

Finally, whether individual embryonic K19[+] TEC have the potential to generate multiple distinct mTEC progeny, or whether heterogeneity in developmental potential lies within the K19[+] population, is not clear. Perhaps relevant to this is our finding that expression of the cell surface marker CD9 might serve as a marker for K19[+] mmTECp. If so, this then excludes the need for K19 detection by cell permeabilization and so will allow for the isolation of K19[+] mTEC progenitors as viable cells for future study. Importantly, further work is required to examine the suitability of CD9 expression, our analysis of TEC heterogeneity using massively parallel flow cytometry may highlight the usefulness of this technology to examine and understand TEC heterogeneity. In sum, our study reveals a population of embryonic thymic epithelial progenitors that gives rise to multiple functionally distinct mTEC populations that sustain the adult thymus medulla. These findings should provide the basis for future work aimed at understanding the development and function of the medulla as a key site for T-cell development and tolerance.

## Methods

### Mice
C57Bl6, RANK-Venus[35], CCL21-tdTom[3], Relb[-/-] (ref. 36) and *Krt19*[CreERT]TdTom mice were bred and maintained at the Biomedical Services Unit at the University of Birmingham under local and national Home Office regulations. *Krt19*[CreERT]TdTom mice were generated by crossing *K19*[CreERT] (ref. 30t; JAX stock #026925) and R26R[LSL]tdTomato[31](JAX stock #007914) strains. To generate embryos, timed matings were set up and vaginal plug detection was designated as embryonic day E0.5. Lineage tracing of K19[+] cells was achieved by a single oral gavage of 4 mg Tamoxifen (VWR) in corn oil (Sigma) to pregnant mice on E12.5 or E15.5 of gestation. A mixture of males and females were used throughout this study.

### Cell isolation
Thymic epithelial cells were isolated from embryonic thymi by digestion using 0.25% Trypsin/0.02% EDTA (Sigma Aldrich). Thymic epithelial cells from neonatal and grafted thymi were isolated by digestion using Collagenase Dispase (2.5 mg/ml Roche) and DNAse-1 (100 mg/ml Roche). All digests were performed at 37′C. Grafted thymi were depleted of CD45[+] cells prior to antibody labelling using anti-CD45 microbeads (Miltenyi) and LS columns (Miltenyi).

### Flow cytometry
The following antibodies were used: CD45-APC eFluor780 (A20, eBioscience, Cat no: 47-0451-82, Lot no: 2375407), EpCAM-BV711 (G8.8 Biolegend, Cat no: 118233, Lot no: B339822), UEA-1 biotin (Vector labs, Cat no: B-1065, Lot no: ZF1204), Ly51 PerCPeF710 (BP-1, eBioscience, Cat no: 46-5891-82, Lot no: 2134423), MHCII AF700 (M5/114.15.2, eBioscience, Cat no: 56-5321-82, Lot no: 2210930), CD80-BV605 (16-10A1, Biolegend, Cat no: 104729, Lot no: B340200), SSEA-1 BV421 (MC-480, Biolegend, Cat no: 125614, Lot not: B311195) Biotinylated antibodies were detected using Streptadvidin-PECy7 (eBioscience, Cat no: 25-4317-82, Lot no: 2034750). Intracellular staining was performed following fixation with 5% formalin solution (Sigma Aldrich) and antibodies used were: Aire AF488 (5H12, Cat no: 53-5934-82, Lot no: 2312434), K19 (EP1580Y, Abcam, Cat no: ab52625,

Lot no: GR3384962-1), DCLK1 (DCAMKL1, Abcam, Cat No: ab31704, Lot no: GR3357375-3), CCL21 (LifeSpan Technologies, Cat no: LS-C104634, Lot no: 55059), SSEA-1-BV421 (MC-480, Biolegend, Cat no: 125614, Lot no: B311195), Relb (Santa Cruz, C-19, Cat no: sc-226, Lot no: F1912). Unconjugated intracellular antibodies were detected using chicken anti-rabbit AF647 (Invitrogen, Cat no: A21443, Lot no: 1881092), or donkey anti-rabbit AF488 (Invitrogen, Cat no: A21206, Lot no: 2072687). All flow cytometric data was acquired using a BD Fortessa and analysed using FlowJo 10.7.1. Gating strategies for flow cytometry analysis can be found in S. Fig. 4.

### Immunofluorescence and microscopy
Embryonic thymic tissue from C57Bl/6 mice was snap-frozen in liquid nitrogen and mounted in OCT prior to cryosectioning. Sections were acetone fixed and stained with the following antibodies: ERTR5 (van Vliet 1985); detected using goat anti-rat IgM AF647 (Invitrogen, Cat no: A21248, Lot no: 2160421), and K19 (EP1580Y, Abcam, Cat no: ab52625, Lot no: GR3384962-1); detected using donkey anti-rabbit IgG AF555 (Invitrogen, Cat no: A31572, Lot no: 2017396). Retention of K19-tdTom in fate-mapping experiments was achieved by fixing thymus tissue in 2% PFA for 2 hours, followed by 20% sucrose overnight, before being snap frozen. Subsequent cryosections were stained with the following antibodies: K5 AF647 (Abcam, EP1601Y Cat no: ab193895, Lot no: GR3416059-2), Aire AF488 (5H12, Cat no: 53-5934-82, Lot no: 2312434), EpCAM1 biotin (G8.8, Biolegend, Cat no: 118204, Lot no: B273843); detected using Streptavidin AF647 (Invitrogen, Cat no: S21374, Lot no: 1990312), DCLK1 (DCAMKL1, Abcam, Cat No: ab31704, Lot no: GR3357375-3); detected using donkey anti-rabbit IgG AF555 (Invitrogen, Cat no: A31572, Lot no: 2017396). Sections were counterstained with DAPI (Invitrogen), and mounted using prolong diamond (Invitrogen). Analysis was performed using a Zeiss LSM 880 confocal microscope and Zeiss Zen Black software.

### Tamoxifen administration
Cre induction in Krt19[CreERT]TdTom embryos was achieved by oral gavage of 4 mg Tamoxifen (VWR) in 200l corn oil (Sigma) to pregnant mice at E12.5 or E15.5 of gestation.

### Thymus transplantation
Freshly isolated lymphoid E13.5 or E16.5 lobes from *Krt19*[CreERT]TdTom mice where tamoxifen had been administered at E12.5 or E15.5 respectively, were transplanted under the kidney capsule of C57Bl/6 WT mice. Grafts were recovered at the equivalent of 4 and 8 weeks post birth and were processed for flow cytometry or immunofluorescence.

### Massively parallel flow cytometry
Experiments were performed as previously described[39]. TEC were isolated from WT E15.5 thymi by enzymatic digestion with Liberase (2.5 mg/ml, Roche) and DNaseI (10 mg/ml, Roche) diluted in PBS (Sigma) at 37 °C. Backbone staining was performed including antibodies directed against CD45 AF700 (30-F11, Biolegend, Cat no: 103128), EpCAM1 PerCPCy5.5 (G8.8, Biolegend, Cat no: 118220), Ly51 PECy7 (6C3, Biolegend, Cat no: 108314), MHCII APC/Fire750 (M5, Biolegend, 107652), CD40 PECy5 (3/12, Biolegend, Cat no: 124617), CD80 BV605 (16-10A1, Biolegend, Cat no: 103128), Sca1 BV510 (D7, Biolegend, Cat no: 108129), Podoplanin BV421 (8.1.1, Biolegend, Cat no: 127423), the *Ulex europaeus* agglutinin I (UEA1, Cy5, Vector laboratories, Cat no: L-1060-5) and Zombie red staining. Subsequently, the stained cells were distributed across the three 96-well plates provided with the LEGENDScreen kit (Biolegend), each well containing a unique PE-labelled exploratory antibody as well as isotype controls and blanks. Due to the low cell numbers obtained only ¼ of the recommended quantity of exploratory antibodies was used. Plates were incubated at 4 °C for 30 min in the dark. Thereafter, fixation was performed using the Cytofix buffer (BD Biosciences) for 1 hour at 4 °C in

the dark. Intracellular staining for K19 AF488 was used as additional backbone marker. The range of exploratory markers was expanded to include FOXN1 PE (2/41, kind gift from Hans-Reimer Rodewald), Aire PE (5H12, Invitrogen, Cat no: 14-5934-82), DCLK1 PE (Abcam Cat no: ab31704), and Ki-67 PE (16A8, Biolegend, Cat no: 652403) which required staining in Cytoperm buffer (BD Biosciences) over-night at 4 °C in the dark. The next day, cells were resuspended in 100 µl FACS buffer before analysis.

**Infinity flow and single-cell clustering and expression analysis**
For the Infinity Flow computational analysis of the LEGENDScreen dataset, the acquired fcs files were gated on CD45⁻EpCAM1⁺ TEC using the FlowJo software. The newly exported fcs files were then used as the dataset for the Infinity Flow pipeline as recently described[38]. The Seurat package was used to further analyse the augmented data matrices generated during this process for hierarchical clustering of the cells and differential expression analysis[46]. Values below zero were set to zero to allow for log normalisation. Markers were filtered by hand to exclude T-cell related and focus on stromal cell related genes.

**Single-cell sequencing analysis**
Gene expression sequencing data were downloaded from GSE182135 generated by Magaletta et al.[26]. Doublets were removed using Doubletfinder and similar quality metrics were used as in the original publication to filter to a final dataset consisting of 52,558 cells. Samples were integrated using mutual nearest neighbour anchor-based analysis in Seurat[47]. Clusters were called using a resolution of 2. Differential expression was performed using FindMarkers in Seurat.

**Statistical analysis**
Prism (GraphPad Software) was used to perform all statistical analyses. Statistical tests used are noted in each figure legend, and $P$ values within each figure. Non-significant differences are not specified. In all figures, bar charts and error bars represent the mean ± SEM, respectively.

**Reporting summary**
Further information on research design is available in the Nature Portfolio Reporting Summary linked to this article.

## Data availability
The authors confirm that data supporting the findings of this study are available in the figures and supplementary figures of the paper. Source data are provided with this paper.

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

## Acknowledgements
We thank staff at the University of Birmingham Biomedical Services Unit for animal husbandry. This work was supported by an MRC Programme Grant to G.A. (MR/T029765/1) and a Wellcome Trust funded Collaborative Award (SynThy, 211944/Z/18/Z) where G.A. and G.A.H. are partners. G.A.H. also received funding from the Swiss National Science Foundation (IZLJZ3_171050; 310030_184672) and the Wellcome Trust (105045/Z/14/Z). A.H. acknowledges the Oxford Health BRC for support. F.K. was supported by a Swiss National Science Foundation Early Postdoc Mobility Fellowship (P2BSP3_188183) and Postdoc Mobility Fellowship (P500PB_206823). I.O. is supported by the JSPS Bilateral Programme (120219928). Y.T. is supported by the Intramural Research Program of the US National Institutes of Health, the National Cancer Institute and the Center for Cancer Research. We thank Hans-Reimer Rodewald for anti-FOXN1 antibody.

## Author contributions

G.A., W.-Y.L., W.E.J. and B.L. conceived the study, and together with F.K. and G.A.H. designed experiments. B.L., A.J.W., W.E.J., S.M.P., F.K., K.D.J., E.J.C. and A.H. performed experiments and analysed data. I.O. and Y.T. provided essential materials and experimental advice. B.L. and G.A. wrote the manuscript with input from all authors.

## Competing interests

The authors declare no competing interests.
