## [Peer Review File · Nature Communications]

Embryonic Keratin19+ Progenitors Generate Multiple Functionally Distinct Progeny To Maintain Epithelial Diversity in The Adult Thymus MedullaREVIEWER COMMENTS

Reviewer #1 (expert in mTEC, AIRE and transcriptional regulation of T cell development):

The manuscript by Lucas et al. examines the potential of embryonic KRT19+ TECs to give rise to specific mature TEC populations into adulthood. The authors show that a subset of embryonic TECs is positive for KRT19 and peaks at E14.5/15.5 when the cortex/medulla areas get structured. These KRT19+ TECs are present at embryonic stages in the cortex and medulla compartments with a higher proportion in UEA1+ Ly51- cells (mTECs) than in UEA1- Ly51+ cells (cTECs). Convincingly, they show that KRT19+ TECs are mostly negative for MHCII and therefore mark immature TECs. The authors performed fate mapping experiments to identify the fate of these cells in newborns. They found staining in AIRE+ mTEChi, Tuft cells and CCL21+ mTEClo.

This work provides interesting new information on a potential embryonic progenitor population positive for KRT19, that would give rise to multiple mTEC subsets crucial to maintain epithelial diversity in the thymic medulla and shape central immune tolerance.

However, important points need to be clarified to support the manuscript conclusions.

1) It is not clear why the authors focused on KRT19 as a marker linked with epithelial progenitors in non-thymic tissues. What are the other candidate markers the authors tested? What is the rational between hepatic progenitor cells (HPC) and thymic epithelial cell progenitors? Why did the authors choose KRT19 and not other HPC markers such as Ascl2 by instance?

2) The pattern of KRT19 expression is not so restricted in the organism, notably during development. Do the authors have data about its expression prior to E12.5? Does KRT19 expression mark thymus formation at E12.5? or is already present at earlier stages such as at the third pharyngeal pouch formation or even earlier at the anterior foregut formation, even at pluripotent stages? scRNAseq datasets of thymus organogenesis are available to help address these questions, such as from Magaletta et al. Nat Comm 2022 (pmid: 35075189).

3) A main caveat of the fate mapping study is that KRT19 turns to be under the control of AIRE in mTEChi of 4-6 wk old mice, with a fold change reaching strong significance. This makes KRT19 unambiguously detectable in mTEChi (cf available public mouse bulk RNAseq datasets). In addition, KRT19 is also widely expressed in Tuft cells (cf available public mouse scRNAseq datasets). Don't the authors think that KRT19 expression in mature mTEChi and Tuft subsets could explain their Tomato staining in newborns after tamoxifen induction at E15.5 when the thymus is composed of a mixture of mature and immature mTECs?

4) KRT19+ TECs are observed at E14.5/E15.5 in the newly formed cortical compartment. These cells are MHCII-. Do the authors know their fate if they don't generate cTEC-lineage cells? Indeed, no Tomato positive cells seem to be detectable in the cortical compartment after birth?

5) The scRNAseq figure lacks labels to identify the clusters. Which clusters correspond to UEA1+Ly51- and UEA1-Ly51+ cells? Can we distinguish MHCIIlo, int and high cells? Do velocity or pseudotime analyses could provide evidence to link KRT19+ TEC fate to existing mTECs and not cTECs at this stage?

Reviewer #2 (expert in thymic stroma and T cell development):

The importance of both thymic epithelial compartments, medulla and cortex, in thymocyte development have been clearly demonstrated in the past decades and the characterization of the diversity of these compartments have considerably progressed recently, notably through scRNAseq analyses. Different embryonic and postnatal progenitors have been proposed to play a role in the persistent generation of TEC compartments, including bipotent progenitors, mainly biased toward cTEC differentiation, and cTEC and mTEC unipotent progenitors. However, their capacity to fully generate the different TEC subsets have not been completely elucidated. In the manuscript

entitled "Embryonic Keratin19+ Progenitors Generate Multiple Functionally Distinct Progeny To Maintain Epithelial Diversity in The Adult Thymus Medulla" by Lucas et al., the authors describe a population of thymic embryonic cells presenting the potential to give rise to distinct thymic epithelial cells (TEC) such as Aire+, CCL21+ or Tuft mTEC cells. Using fate-mapping approach, the authors showed the persistence of mature TEC derived from these Keratin 19 (K19) expressing progenitors in the adulthood. While the understanding of thymus development and in particular the characterization of progenitors able to (re)generate thymic compartment is of importance, notably as it could open therapeutic perspectives, several points need to be addressed to fully understand the characteristics and the potential of these K19+ TEC.

General comments:

First, the relative importance of the K19+ progenitor population in the generation of TEC and in particular mTEC is difficult to assess in this study. Indeed, in fate-mapping experiments only a small portion of K19+ cells expressed the tdTomato marker (10% 24h after Tamoxifen treatment) and as such the quantification of the progeny of K19+ cells is underestimated. Moreover, no quantifications of the percentage of fate-mapped cells in neonate or adult is provided.

Furthermore, while self-renewal is one of the key features of progenitor cells, it is difficult to firmly conclude on the renewal capacity of the K19+ progenitor cells. Clonogenic experiments of sorted tdTomato expressing cells or CD9+ TEC would help to further assess this particular parameter. Indeed, the experiments presented here demonstrated only the potential of K19+ cells to differentiate in different mTEC sub-populations which persist for 4 weeks. Half-life of TEC cells should also be discussed.

Finally, it would have been important to assess the expression of already described markers of mTEC progenitor populations such as SSEA1 or claudin 3 and 4, for example, in the K19 population. This would allow to determine the relationship of this heterogeneous K19+ population with the already described progenitors.

In a second point, while the authors mainly focus on the mTEC differentiation potentials of this population, different results underline a potential role of the K19+ progenitors in the generation of cTEC cells. Indeed, at E12,5 30% of K19+ TEC cells express the LY51 cTEC marker, as compared to 3% in K19- TEC population. At E15,5 this percentage remains stable in K19+ cells while the expression of the mTEC marker UEA1 increases drastically. Moreover, in the fate-mapping experiments, cTEC cells are also generated, albeit at a lower percentage (5%). As such it would be important to better characterize these cTEC cells, to discuss the kinetic of their generation and their localization as no fate-mapped cells have been observed in the cortex compartment at birth, while K19+ cells are found in all compartments at E15,5.

Specific points:

The figure quality is really poor and need to be increased. In particular, the signs + and - are not readable.

Line 89-90: the sentence is not finished.

The authors mentioned that they investigated expression of different putative progenitor markers to identify epithelial progenitors. The data are supposed to be in supplementary figure 1 which shows only a gating strategy of TEC.

Results of figure 3 have already been published and could be presented as a supplementary figure.

In the figure 4A, the gating of Ly51+ population in the first "Total Epcam+" plot is different from the gating of the same population in the next 2 plots. This need to be corrected as results are supposed to come from the same analyzed tube.

The Ly51+UEA1- histogram is the same in figure 4C and figure 3D E15.5. Please show a different plot.

In figure 5A, the quantification of the % of tdTomato+ cells needs to be depicted and for a better interpretation of the results, the % of K19+ cells and their relative expression of Ly51, UEA1 should be shown at PNd0. The authors should also discuss the difference in percentage of tdTomato+ at E16,5 (1d after tamoxifen treatment) and at PNd0.

In figure 6, the percentage of tdTomato in EpCAM+ cells and their relative expression of Ly51 and UEA1 needs to be depicted to a full interpretation of the results. The authors should also discuss the difference in the percentage of mTEChi between figure 5 and 6.

Line 181, the authors should refer to figure 4 instead of figure 3

In Figure 10, the authors used a different digestion protocol. As it could influence the isolation of TEC populations, they should explain this choice and show evidence of similar results with both digestion protocols

Reviewer #3 (expert in massively parallel flow cytometry):

Summary: This study by Lucas et al. utilizes sophisticated ontogenetic and inducible fate-mapping tools in order to identify a precursor of mTEC in murine Thymus. They define these newly identified progenitors as being K19+ and provide data supporting that these cells can differentiate into multiple mTEC lineages within the Thymus. For the purposes of this review – I am restricting my comments to the aspects relating to high-dimensional flow cytometry approaches as I have background in this area but not in thymic development.

1) After identifying the K19+ TEC (mmTECp) as a progenitor population utilizing genetic fate tracking tools (KrtCreERT2-mice) the authors sought to establish additional surface markers suitable for identification of this population in WT mice where fate tracking with this approach is not possible. The authors utilize massively parallel flow cytometry coupled with a machine learning-based algorithm (Infinity Flow) to identify markers of K19+ TEC in single cell suspension of WT murine thymus. The utilization of Infinity Flow for this purpose appears to be an appropriate approach and the technique has been properly executed.

2) Using the above approach CD9 is identified as a marker that nicely correlates with K19 expression and suggested to be an alternate marker for the mmTECp population. No data is shown however to cross-validate this beyond a simply K19+ stain. This validation is necessary in order to truly interpret the caliber of CD9 as a marker – suggestions to accomplish this would be 1) Assess CD9 expression in the context of the K19-cre fate tracking system to ensure the populations indeed mark the expected subset. 2) Perform comparative transcriptomic analysis sorting on CD9+ TEC vs K19+ TEC in order to ensure that the two markers truly identify overlapping populations.

If one or more of the above limitations are addressed I believe the data presented in this area is sufficiently rigorous and interesting to merit publication. Again noting I've only provided direct assessment of the High Dimensional aspects of this analysis.

RESPONSE TO REVIEWERS' COMMENTS

Reviewer 1.

We thank the reviewer for their positive comments and support, who indicates our findings are 'convincing' and provide 'interesting new information' on mTEC progenitors crucial to central tolerance.

- 1. It is not clear why the authors focused on KRT19 as a marker linked with epithelial progenitors in non-thymic tissues. What are the other candidate markers the authors tested? What is the rationale between hepatic progenitor cells (HPC) and thymic epithelial cell progenitors? Why did the authors choose KRT19 and not other HPC markers such as Ascl2 by instance?**

The reason for our focus on K19 in the context of thymus biology initially arose through discussions with Dr. Wei-Yu Lu, an Institute member who previously showed that K19 expression identifies epithelial progenitors in liver (e.g. Nature Cell Biol. 2015). From his data, and also other interesting comparisons that include the endodermal origin of both thymus and liver, we formed the hypothesis that K19 might also be linked to epithelial progenitors in thymus. To our knowledge, this has never previously been tested. Dr. Lu's inducible fate mapping approaches that were available locally made testing this hypothesis practically possible. This is now described further on pages 4 and 5.

- 2. The pattern of KRT19 expression is not so restricted in the organism, notably during development. Do the authors have data about its expression prior to E12.5? Does KRT19 expression mark thymus formation at E12.5? or is already present at earlier stages such as at the third pharyngeal pouch formation or even earlier at the anterior foregut formation, even at pluripotent stages? scRNAseq datasets of thymus organogenesis are available to help address these questions, such as from Magaletta et al. Nat Comm 2022 (pmid: 35075189).**

We agree that analysis of K19 expression prior to E12.5 is interesting and could provide further important information in relation to early stages of thymus organogenesis. As suggested, we have analysed the published data set of Magaletta et al for K19 expression. Interestingly, we find that K19 expression is detectable in TEC at E9.5 and E10.5 of gestation, as well as in the 1st and 2nd pharyngeal pouches, and the parathyroid and thyroid. This data is now shown in Supp. Figure 3, and discussed on page 5.

- 3. A main caveat of the fate mapping study is that KRT19 turns to be under the control of AIRE in mTEC^{hi} of 4-6 wk old mice, with a fold change reaching strong significance. This makes KRT19 unambiguously detectable in mTEC^{hi} (cf available public mouse bulk RNAseq datasets). In addition, KRT19 is also widely expressed in Tuft cells (cf available public mouse scRNAseq datasets). Don't the authors think that KRT19 expression in mature mTEC^{hi} and Tuft subsets could explain their Tomato staining in newborns after tamoxifen induction at E15.5 when the thymus is composed of a mixture of mature and immature mTECs?**

The reviewer refers to patterns of K19 mRNA expression in mature mTEC of adult mice obtained from public RNAseq datasets, and asks whether our data showing K19-mediated fate mapping of tuft cells and mature mTEC is caused by their K19 expression. It is important to note that in our inducible K19 fate mapping experiments, administering a single injection of Tamoxifen at E15.5 then limits Cre recombination to a single day of embryogenesis. Given that tuft cells are not yet present at this timepoint, (first detectable after birth, Lucas et al Nature Comms 2020), pre-labelling of tuft cells at the time of Tamoxifen injection cannot explain our detection of fate mapped tuft cells. Moreover, new data included in Fig 3b shows that embryonic Aire⁺ cells lack K19 expression, and so could not have been fate mapped by K19^{Cre} as pre-existing Aire⁺ cells. Consequently, labelling of Aire⁺ cells at the time of Tamoxifen injection cannot explain our detection of fate

mapped Aire⁺ cells. Thus, although it is interesting that K19 is expressed by mature mTEC in the adult thymus, we exclude these cells from our fate mapping studies by limiting induction of fate mapping to a very short period of embryonic life, where K19⁺ cells do not contain mature Aire⁺ mTEC and tuft cells. This is explained on page 8.

4. KRT19⁺ TECs are observed at E14.5/E15.5 in the newly formed cortical compartment. These cells are MHCII⁻. Do the authors know their fate if they don't generate cTEC-lineage cells? Indeed, no Tomato positive cells seem to be detectable in the cortical compartment after birth?

The reviewer raises an interesting point in relation to data shown in Figure 3, where some K19⁺ TEC have a UEA1⁻Ly51⁺ phenotype, yet Tomato⁺ cells are undetectable in cTEC after birth. More generally, this point relates to the possible cTEC potential of embryonic K19⁺ cells. Interestingly, in the embryonic thymus, cells expressing a cTEC phenotype have been shown to give rise to both cTEC and mTEC, and so have been termed 'cTEC-like cells'. Thus, one possibility for the presence of K19⁺ cells with a cTEC phenotype in the embryo, but the absence of K19-fate mapped cells after birth, is that embryonic K19⁺Ly51⁺ cells are in fact cTEC-like cells with mTEC potential, and not mature cTEC. Alternatively, K19⁺ cells may retain some residual cTEC potential alongside their strong bias to mTEC generation. Indeed, we now show that the small (5%) proportion of fate mapped cells displaying an Ly51⁺UEA1⁻ phenotype contain both MHCII⁻ and MHCII⁺ cells, which may be evidence for both possibilities. We discuss these possibilities in detail on pages 14 and 15.

5. The scRNAseq figure lacks labels to identify the clusters. Which clusters correspond to UEA1⁺Ly51⁻ and UEA1⁻Ly51⁺ cells? Can we distinguish MHCII^{lo}, int and high cells? Do velocity or pseudotime analyses could provide evidence to link KRT19⁺ TEC fate to existing mTECs and not cTECs at this stage?

This comment is in relation to data shown in Figure 9. It is important to clarify that all data shown in this Figure is on protein expression detected by Infinity Flow analysis, and not scRNAseq data. We labelled clusters in Fig 9A numerically rather than use TEC subset names, because most E15.5 TEC are still immature, and their relationship to well defined subsets present in adult thymus (eg mature cTEC, tuft cells) is not clear. Importantly however, the reviewer suggests further analysis of three markers in relation to clustering (UEA1 and Ly51 and MHCII). We now include UMAPS and violin plots to provide further information on their patterns of expression, and discuss these points on page 12 and 13.

Reviewer 2.

We thank the reviewer for emphasizing the importance of our study in relation to understanding thymus development and the opportunity studies of this kind provide to therapeutic perspectives.

General Comments.

1. First, the relative importance of the K19⁺ progenitor population in the generation of TEC and in particular mTEC is difficult to assess in this study. Indeed, in fate-mapping experiments only a small portion of K19⁺ cells expressed the tdTomato marker (10% 24h after Tamoxifen treatment) and as such the quantification of the progeny of K19⁺ cells is underestimated. Moreover, no quantifications of the percentage of fate-mapped cells in neonate or adult is provided.

This issue relates to the frequency of mTEC that are labelled during inducible K19^{Cre} fate mapping experiments, and the importance the contribution of K19⁺ cells to mTEC development. The reviewer is correct that our experiments involve successful fate mapping of a portion K19⁺ cells, we also agree that this then likely underestimates the contribution of K19⁺ cells to the mTEC population. However, it is important to point out that this portion is sufficiently large to track up to 8 weeks post labelling, and allows us to make our main conclusion regarding heterogeneity within the progeny of fate mapped cells. Given that inducible fate mapping labels only a proportion of K19⁺ cells, we cannot, and do not, make conclusions on whether all mTEC are exclusively derived

from K19⁺ cells. As with all fate mapping experiments, we can only comment on the cells that are labelled, and the progeny of these labelled cells. To clarify this, we now include discussion on pages 14 and 15 to indicate the possibility that diverse mTEC may also be generated from K19⁻ cells. As requested, we also now provide data quantitating the percentages of fate mapped cells in all fate mapping experiments (Figure 4A, Figure 5B, Figure 7B).

- 2. Furthermore, while self-renewal is one of the key features of progenitor cells, it is difficult to firmly conclude on the renewal capacity of the K19⁺ progenitor cells. Clonogenic experiments of sorted tdTomato expressing cells or CD9⁺ TEC would help to further assess this particular parameter. Indeed, the experiments presented here demonstrated only the potential of K19⁺ cells to differentiate in different mTEC sub-populations which persist for 4 weeks. Half-life of TEC cells should also be discussed.**
- 3. Finally, it would have been important to assess the expression of already described markers of mTEC progenitor populations such as SSEA1 or claudin 3 and 4, for example, in the K19 population. This would allow to determine the relationship of this heterogenous K19⁺ population with the already described progenitors.**

We would like to respond to both of these points together. They focus on how K19⁺ TEC progenitors described here may relate to SSEA1⁺ mTEC stem cells described by Sekai et al 2014. We cited and discussed this paper in our original manuscript. It is important to emphasise the key aim of our study was to examine the developmental origins of functionally distinct TEC populations. From this, our key finding is that the embryonic thymus contains a TEC subset, defined by K19 expression, that gives rise to multiple functionally distinct mTEC subsets including Aire⁺ cells, tuft cells and CCL21⁺ cells. To our knowledge, a shared origin of these cells has not yet been reported. While we feel TEC stem cell/self-renewal is certainly of interest, given the focus of our study we were careful not to include these terms to describe our data on K19⁺ progenitors, and we feel such analysis lies outside of the scope of our current study. However, to aid in relating our findings to the issue of mTEC stem cells, we now include longer-term analysis of embryonically fate mapped K19⁺ cells, where we show fate mapped cells continue to persist as multiple functionally distinct mTEC subsets at the equivalent of 8 weeks of age (Figure 5F-I). As suggested, we also now include new data on SSEA1 expression, and show that approximately half of the SSEA1⁺ cells at E12.5 and E15.5 express K19 (Figure 5J, 5K). We discuss these recent findings in relation to the earlier findings of Sekai et al on pages 9, 10 and 16, 17. We also cite additional publications from the Hamazaki lab on the issue of mTEC stem cells (References 33 and 34).

- 4. In a second point, while the authors mainly focus on the mTEC differentiation potentials of this population, different results underline a potential role of the K19⁺ progenitors in the generation of cTEC cells. Indeed, at E12,5 30% of K19⁺ TEC cells express the LY51 cTEC marker, as compared to 3% in K19⁻ TEC population. At E15,5 this percentage remains stable in K19⁺ cells while the expression of the mTEC marker UEA1 increases drastically. Moreover, in the fate-mapping experiments, cTEC cells are also generated, albeit at a lower percentage (5%). As such it would be important to better characterize these cTEC cells, to discuss the kinetic of their generation and their localization as no fate-mapped cells have been observed in the cortex compartment at birth, while K19⁺ cells are found in all compartments at E15,5.**

This point is similar to that of reviewer 1, point 4 and relates to the issue of the cTEC lineage in relation to K19 expression by embryonic TEC. The reviewer points out that we show some K19⁺ embryonic TEC express the cTEC marker Ly51 and in fate mapping experiments we find a small population of fate mapped cells with an Ly51⁺UEA1⁻ phenotype. However, and as discussed in reviewer 1 point 4, cells with this cTEC phenotype could include both immature 'cTEC-like' cells that have both cTEC and mTEC potential, as well as mature cTEC. For this reason, throughout our study we did not refer to K19⁺ cells as being 'mTEC committed', allowing for the possibility that they may contain residual cTEC potential. On this point, and as requested, we have further characterized these cells in relation to MHCII expression. We now show that fate mapped K19⁺

cells with an Ly51⁺UEA1⁻ phenotype are a mixture of both MHCII^{neg} and MHCII⁺ cells (Figure 4G). The possibility that the small percentage of Ly51⁺ K19 fate mapped cells then reflects both residual cTEC potential and immature cTEC-like cells is discussed on page 14-15.

Specific Points.

- 1. The figure quality is really poor and need to be increased. In particular, the signs + and – are not readable.**

High quality images at the appropriate resolution have now been provided.

- 2. Line 89-90: the sentence is not finished.**

This has now been corrected.

- 3. The authors mentioned that they investigated expression of different putative progenitor markers to identify epithelial progenitors. The data are supposed to be in supplementary Figure 1 which shows only a gating strategy of TEC.**

We apologise for this confusion, likely caused by inaccurate syntax in this sentence of the manuscript. This has now been rewritten (page 4-5) to include our ideas behind the hypothesis that K19 expression might be a useful tool to study TEC development. See also point 1, reviewer 1.

- 4. Results of Figure 3 have already been published and could be presented as a Supplementary Figure.**

As requested, this data is now included as Supplementary Figure 1.

- 5. In the figure 4A, the gating of Ly51+ population in the first “Total Epcam+” plot is different from the gating of the same population in the next 2 plots. This need to be corrected as results are supposed to come from the same analyzed tube.**

As requested, new gates have been included in the Figure.

- 6. The Ly51+UEA1- histogram is the same in figure 4C and figure 3D E15.5. Please show a different plot.**

As requested, a new plots has been included (See Supp Figure 1D).

- 7. In figure 5A, the quantification of the % of tdTomato+ cells needs to be depicted and for a better interpretation of the results, the % of K19+ cells and their relative expression of Ly51, UEA1 should be shown at PNd0. The authors should also discuss the difference in percentage of tdTomato+ at E16,5 (1d after tamoxifen treatment) and at PNd0.**

Data showing quantification of tdTomato⁺ cells is now included (now Figure 4A), as well as their expression of Ly51 and UEA1 (now Figure 4B, C). We also discuss the percentages of tdTomato⁺ cells at E16.5 and PNd0 on page 7.

- 8. In figure 6, the percentage of tdTomato in EpCAM+ cells and their relative expression of Ly51 and UEA1 needs to be depicted to a full interpretation of the results. authors should also discuss the difference in the percentage of mTEChi between figure 5 and 6.**

As requested, we now include data showing the percentage of EpCAM1⁺ cells that are tdTomato⁺ (now Figure 5B), and their expression of Ly51 and UEA1 (now Figure 5C). We also discuss the percentage of mTEC^{hi} TEC in fate mapping at birth and 4 weeks post-birth (now Figures 4 and 5) on pages 8 and 9.

- 9. Line 181, the authors should refer to figure 4 instead of figure 3.**

We have now corrected references to all Figures that correlates with the new and revised Figure order.

10. In Figure 10, the authors used a different digestion protocol. As it could influence the isolation of TEC populations, they should explain this choice and show evidence of similar results with both digestion protocol.

The reasons we chose Liberase as part of the digestion protocol for the Infinity Flow analysis was to try and prevent loss of cell surface proteins being analysed. This is explained on page 11. Use of this liberase protocol for Infinity Flow meant we used the same protocol established in our companion manuscript from Klein et al. Furthermore, it is important to note that while Affinity Flow data in our study provides information on TEC phenotyping, we not draw direct comparisons between this data and our ontogenetic and fate mapping data, where the same enzymatic digestion is used throughout.

Reviewer 3.

We thank this reviewer for their support on our use of ‘sophisticated ontogenetic and inducible fate mapping tools to identify a precursor of mTEC in murine thymus’. Their comments relate to our use of Infinity Flow to further define K19⁺ mTEC progenitors.

- 1) After identifying the K19⁺ TEC (mmTECp) as a progenitor population utilizing genetic fate tracking tools (KrtCreERT2-mice) the authors sought to establish additional surface markers suitable for identification of this population in WT mice where fate tracking with this approach is not possible. The authors utilize massively parallel flow cytometry coupled with a machine learning-based algorithm (Infinity Flow) to identify markers of K19⁺ TEC in single cell suspension of WT murine thymus. The utilization of Infinity Flow for this purpose appears to be an appropriate approach and the technique has been properly executed.**

We thank the reviewer for highlighting the relevance and appropriateness of Infinity Flow to further define K19⁺ mTEC progenitors, and also our proper execution of this technique.

- 2) Using the above approach CD9 is identified as a marker that nicely correlates with K19 expression and suggested to be an alternate marker for the mmTECp population. No data is shown however to cross-validate this beyond a simply K19⁺ stain. This validation is necessary in order to truly interpret the caliber of CD9 as a marker – suggestions to accomplish this would be 1) Assess CD9 expression in the context of the K19-cre fate tracking system to ensure the populations indeed mark the expected subset. 2) Perform comparative transcriptomic analysis sorting on CD9⁺ TEC vs K19⁺ TEC in order to ensure that the two markers truly identify overlapping populations.**

Specifically, the reviewer is interested in our finding on CD9 expression in relation to K19⁺ progenitors, and suggests approaches to specifically test whether CD9⁺ cells behave in the same way as K19⁺ cells. Here, it is important to emphasise the reasons for Infinity Flow analysis in our study, First, we wanted to demonstrate the potential of this technique to the TEC field, and this fits well with, and extends, findings in the companion manuscript by Klein et al. Second, we aimed to provide initial data on the detailed phenotypic profile of K19⁺ cells that would be of use in future studies. Amongst the many markers that were found to be present or absent in K19⁺ cells, we provided some additional analysis on CD9 expression. Importantly, we did not want to give the impression that our data demonstrates that CD9⁺ cells are the same as K19⁺ cells, simply that CD9 may be an interesting marker to help in future studies on mTEC progenitors. We apologise for any unclear comments in the manuscript on this. We now clarify our use of Infinity Flow, and the findings on CD9 expression, on page 12 and 13. In addition, we also refer to point 5 from reviewer 1, where further analysis of Infinity Flow data is now included in Fig 9.

In summary, we thank the reviewers for their comments, which have allowed us to include new data and provide further discussion/interpretation based their suggestions.

REVIEWERS' COMMENTS

Reviewer #1 (expert in mTEC, AIRE and transcriptional regulation of T cell development):

The authors have addressed all my concerns. They substantially clarified key points and performed required additional analyses improving the quality of the manuscript. The concept of K19+ cTEC-like cells with mTEC potential (mmTECp) is appealing and strengthened in this new version.

Reviewer #2 (expert in thymic stroma and T cell development):

The quality of the manuscript has improved following the revisions and the majority of the reviewers' comments have been addressed.

Reviewer #3 (expert in massively parallel flow cytometry):

Thank you to the authors for their response to my minor critiques. Based on the additional information provided in the text and figure with respect to the intended interpretation of the Infinity Flow data provided in Figure 9 (specifically regarding validation of how CD9 expression fits within the overall model.) I've no remaining issues with respect to the way the data was collected and presented.

One minor note I will make just for Author consideration. It remains a bit unclear to me what value the inclusion of the CD9 finding brings to the story. I do see value in the Infinity Flow experiment and the comprehensive profiling you've done (particularly in light of including some additional analysis in the revised manuscript), however, without additional context (which is admittedly out of scope for your study) the CD9 finding doesn't seem to quite fit into this story for me. It isn't a significant detractor but it is challenging to interpret whether there is much meaning to the CD9 expression pattern in its current state. It may simply be a better fit in a future story where more depth and context can be provided.

RESPONSE TO REVIEWERS' COMMENTS

Reviewer 3.

One minor note I will make just for Author consideration. It remains a bit unclear to me what value the inclusion of the CD9 finding brings to the story. I do see value in the Infinity Flow experiment and the comprehensive profiling you've done (particularly in light of including some additional analysis in the revised manuscript), however, without additional context (which is admittedly out of scope for your study) the CD9 finding doesn't seem to quite fit into this story for me. It isn't a significant detractor but it is challenging to interpret whether there is much meaning to the CD9 expression pattern in its current state. It may simply be a better fit in a future story where more depth and context can be provided.

The reviewer discusses the relevance of including the Infinity Flow data in our manuscript. We believe that including this data is important 1) as it may help identify phenotypic markers to study K19⁺ mTEC progenitors in wildtype mice and 2) it demonstrates the usefulness of this approach to study thymic epithelial cells. For these reasons, we think including this data is valuable to our study. This is highlighted on page 16.

In summary, we thank the reviewers for their comments, which have allowed us to include new data and provide further discussion/interpretation based their suggestions.